# Massively parallel screen uncovers many rare 3′ UTR variants regulating mRNA abundance of cancer driver genes

Ting Fu [1,2,8], Kofi Amoah [2,3,8], Tracey W. Chan[2,3], Jae Hoon Bahn[2], Jae-Hyung Lee [2,4], Sari Terrazas[2,5], Rockie Chong[6], Sriram Kosuri[6] & Xinshu Xiao [1,2,3,5,7] ✉

Understanding the function of rare non-coding variants represents a significant challenge. Using MapUTR, a screening method, we studied the function of rare 3′ UTR variants affecting mRNA abundance post-transcriptionally. Among 17,301 rare gnomAD variants, an average of 24.5% were functional, with 70% in cancer-related genes, many in critical cancer pathways. This observation motivated an interrogation of 11,929 somatic mutations, uncovering 3928 (33%) functional mutations in 155 cancer driver genes. Functional MapUTR variants were enriched in microRNA- or protein-binding sites and may underlie outlier gene expression in tumors. Further, we introduce untranslated tumor mutational burden (uTMB), a metric reflecting the amount of somatic functional MapUTR variants of a tumor and show its potential in predicting patient survival. Through prime editing, we characterized three variants in cancer-relevant genes (*MFN2*, *FOSL2*, and *IRAK1*), demonstrating their cancer-driving potential. Our study elucidates the function of tens of thousands of non-coding variants, nominates non-coding cancer driver mutations, and demonstrates their potential contributions to cancer.

Whole-genome sequencing efforts have revealed the landscape of human non-coding genetic variants[1]. It is increasingly recognized that non-coding variants contribute significantly to human diseases, including cancer[2–4]. Despite the substantial progress toward defining functional elements in the non-coding genome, a primary challenge in human genetics is to pinpoint the biological mechanisms through which non-coding genetic variants confer disease risks. Genome-wide association studies (GWASs) have been conducted extensively to detect disease-associated genomic loci. However, only a small proportion of disease heritability has been explained[5]. This is partly due to

the limited statistical power of GWASs and their reliance on arrays of common single-nucleotide polymorphisms (SNPs), which makes it difficult to detect rare variants with modest effects[5,6].

At the population level, an overwhelming majority of genetic variants are rare[1,7]. In contrast to common variants, which typically have small, additive effects on complex traits, rare variants can have large functional effects[8,9]. It is hypothesized that rare variants contribute substantially to the missing heritability of complex traits and diseases[10–13]. For example, recent studies using whole-genome sequencing data have successfully uncovered much of the heritability

[1]Molecular, Cellular and Integrative Physiology Interdepartmental Program, University of California, Los Angeles, Los Angeles, CA 90095, USA. [2]Department of Integrative Biology and Physiology, University of California, Los Angeles, Los Angeles, CA 90095, USA. [3]Bioinformatics Interdepartmental Program, University of California, Los Angeles, Los Angeles, CA 90095, USA. [4]Department of Life and Nanopharmaceutical Sciences & Oral Microbiology, School of Dentistry, Kyung Hee University, Seoul, South Korea. [5]Molecular Biology Interdepartmental Program, University of California, Los Angeles, Los Angeles, CA 90095, USA. [6]Department of Chemistry and Biochemistry, University of California, Los Angeles, Los Angeles, CA 90095, USA. [7]Jonsson Comprehensive Cancer Center, University of California, Los Angeles, Los Angeles, CA 90095, USA. [8]These authors contributed equally: Ting Fu, Kofi Amoah. ✉e-mail: gxxiao@ucla.edu

of complex traits such as height, primarily through the detection of rare variants[13]. However, due to their low frequencies, rare variants are often outside the reach of most statistically powered association studies, despite the increasing recognition of their disease implications[14–16]. Thus, decoding the functions of non-coding rare variants will greatly inform a better understanding of disease mechanisms.

In general, non-coding variants may affect many aspects of gene regulation, including both transcriptional and post-transcriptional processes. It was estimated that about 60% of the variation in protein expression arises from post-transcriptional regulation[17]. However, in contrast to the substantial progress made in understanding transcriptional regulation, pinpointing functional variants in post-transcriptional steps, for example, via regulation by the 3′ untranslated regions (UTRs), remains a major challenge[9,18].

The 3′ UTR is a primary hub of gene regulation, affecting many critical processes such as RNA stability, RNA localization and translation[19]. This region harbors numerous *cis*-regulatory elements, interacting with *trans*-factors including microRNAs (miRNAs) and RNA-binding proteins (RBPs)[20]. For genetic variants in the 3′ UTRs, a primary mode of function is by disrupting the *cis*-regulatory elements and affecting the binding of *trans*-acting factors[21]. Expression quantitative trait loci (eQTL) studies revealed a significant enrichment of putative functional non-coding variants in the 3′ UTRs, often representing the largest enrichment among all types of non-coding regions[22–24].

In addition to the global association analysis of eQTLs, massively parallel reporter assays (MPRAs) are emerging as effective means to decipher 3′ UTR function[18,25–29]. It possesses unique advantages as a high-throughput functional assay that allows the nomination of causal variants in non-coding regions[18,25,29]. However, previous eQTL and MPRA studies primarily focused on the functions of common genetic variants[18,23,25,29]. Little attention has been given to rare variants (e.g., < 1% in Minor Allele Frequency), which constitute the majority of 3′ UTR variants with unknown function[1,7]. The recent reports of example rare functional 3′ UTR variants highlight the need of global analyses of such variants for a more complete understanding of the genetic basis of disease[18,30,31].

In this study, using our developed assay, a massively parallel screen for rare 3′ UTR variants (MapUTR), we tested the function of 17,301 rare variants in regulating mRNA abundance in two human cell lines. We observed that 70% of functional rare variants were in cancer-relevant genes. Thus, we further tested a special type of rare variants, cancer somatic mutations, in well-known cancer driver genes. Our data demonstrate the likely existence of abundant 3′ UTR cancer mutations in cancer driver genes. In addition, functional MapUTR variants enabled the definition of a metric, untranslated tumor mutational burden (uTMB), which has the potential to predict patient survival.

## Results

### The MapUTR method to identify functional 3′ UTR variants regulating mRNA abundance

In MapUTR, we cloned the synthetic DNA oligonucleotides (oligos) containing 3′ UTR variants and their flanking sequences (158–164 nucleotides (nt) in total) into the 3′ UTR of the *eGFP* gene (Fig. 1a, Methods). This reporter gene is driven by a strong promoter, the CMV early enhancer/chicken beta actin (*CAG*) promoter[32], which allows the identification of functional variants primarily affecting post-transcriptional (rather than transcriptional) regulation. The plasmid library was electroporated into HEK293 or HeLa cells to test for mRNA abundance (Fig. 1a).

To avoid exhaustion of cellular machineries in the episomal reporter assay, it is expected that minimizing the amount of transfected DNA is beneficial. We thus first determined the minimum amount of DNA per cell (DNA/Cell ratio) for MapUTR. For three known functional variants in the literature[25,33], we measured their impacts given three DNA/Cell ratios respectively. The largest ratio, 4ug DNA per 1 million cells (4 μg/1 M), was chosen as it was used in previous literature[34] and the smallest ratio, 40 ng/1 M, was chosen as lower ratios did not permit recovery of the RNA libraries. We observed that the smaller DNA/Cell ratio yielded larger effect sizes, in line with the expected directions of RNA/DNA ratios (Supplementary Fig. 1). We hypothesized that this increased sensitivity may be due to the low DNA input, which avoids exhaustion of the cellular machineries. However, we found that the targeted RNA-seq library had very low yield (< 1 ng) at 40 ng/1 M DNA/Cell ratio. Thus, we opted to use a DNA/Cell ratio (200 ng/1 M) that is as low as possible while yielding expected directions of RNA/DNA ratios (Supplementary Fig. 1) and reliable generation of RNA-seq libraries (10–180 ng). Compared to other studies[18,34,35], we were able to lower the DNA/Cell input ratio by 5–20 fold.

After electroporation of the plasmid library, total RNA was extracted for RNA-seq library generation targeting the tested 3′ UTRs (Fig. 1a, Supplementary Fig. 2a). A DNA-seq library is also needed to allow for normalizations of RNA expression. To this end, either pre- or post-transfection DNA-seq was used in previous studies[18,25,34–36]. To determine our protocol, we collected both pre- and post-transfection DNA-seq libraries with three biological replicates each, incorporating 15-mer unique molecular identifiers (UMIs) in library generation (Supplementary Fig. 2b). Following sequencing, we calculated the allele frequency (AF) (alt/(alt+ref)) for each variant using the UMI counts from the two types of DNA-seq libraries. Indeed, their AFs are highly correlated (Supplementary Fig. 3a), especially at high UMI abundances, although the correlation is still significant given low UMI counts (1st quantile) (Supplementary Fig. 3b, c). Given the ease of generating pre-transfection DNA libraries, we opted to use pre-transfection DNA to perform RNA abundance normalization.

UMIs were also incorporated into RNA-seq libraries (Supplementary Fig. 2a). In the data analysis steps, UMIs in the DNA-seq and RNA-seq data were extracted to enable removal of PCR duplicates, followed by read alignment, data normalization and detection of functional variants (Fig. 1b, Methods). To measure the impact of each allele on mRNA abundance, we calculated an "activity score" as the normalized relative read number (RNA/DNA) for that allele (Fig. 1c, Methods). We further compared the activity scores of the alternative and reference alleles to reach at a "relative activity score" (Fig. 1c, Methods).

### MapUTR captures functional effects of random mutations within known *cis*-regulatory elements in the 3′ UTR

To test the performance of MapUTR, we picked 5 known 3′ UTR motifs (Supplementary Table 1) reported in the literature[25] that alter mRNA stability. We mutated each base respectively within the motif and its surrounding regions (22–23 nt on each side, Fig. 1d). For each base, all 3 possible alternative alleles were included individually. The resulting pool of oligo sequences was tested with MapUTR in both HEK293 and HeLa cells. For all 5 motifs, we observed a low mismatch rate relative to the reference sequence in the DNA-seq data (0.057% on average) outside of the mutated regions (Fig. 1e, Supplementary Fig. 4). This observation indicates that the oligo synthesis and subsequent experimental steps produced generally low error rates.

Next, we compared the impact of each mutation in the known motifs and their surrounding sequences on mRNA abundance. As shown in Fig. 1f, mutations in the functional motifs induced considerable changes in relative mRNA expression, whereas the flanking regions were associated with relatively small mutation-induced variations. In addition, the alternative alleles induced an overall increase in mRNA abundance (positive relative activity scores), consistent with the known role in mRNA destabilization of each motif[25]. Our results support the effectiveness of MapUTR in capturing biologically relevant post-transcriptional regulatory events.

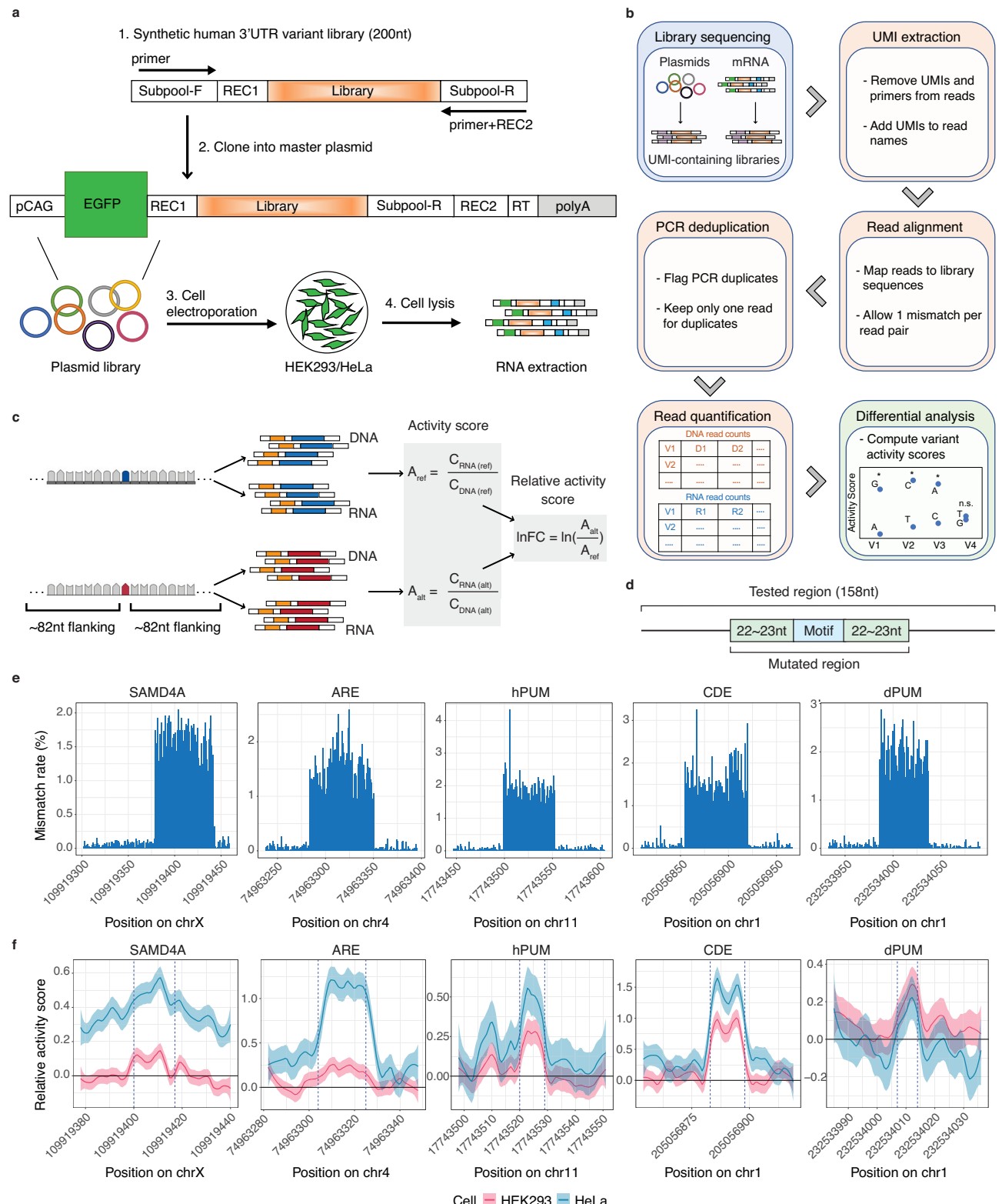

## Identification of functional rare 3′ UTR variants with MapUTR

We next applied MapUTR to test the functionality of rare genetic variants in the 3′ UTR. From the Genome Aggregation Database (gnomAD)[37], we extracted rare 3′ UTR variants defined as those with an adjusted minor allele frequency (adjAF) < 0.01. After removing sequences incompatible with the cloning strategy (e.g., shared similarity with restriction enzyme sites or primer sequences, see Methods and Supplementary Table 2), we selected 17,301 variants to be tested with MapUTR, 1044 of which were also reported in clinically relevant databases (ClinVar[38], CIViC[39], COSMIC[40], iGAP[41]).

We synthesized 200nt-long oligos harboring each rare variant and its flanking region, together with subpool primers and restriction sites (Methods, Fig. 1a,c). A relatively low error rate (average 0.016%) was again observed in the DNA-seq and RNA-seq reads (Fig. 2a). The average error rate across each designed sequence or at the vicinity

**Fig. 1 | MapUTR captures functional 3′ UTR variants in known functional motifs. a** Experimental Design of MapUTR. Subpool-F and Subpool-R represent the 15-mer primer sequences to amplify the oligo pool. REC1 and REC2 represent restriction enzyme sites. pCAG, the CMV early enhancer/chicken beta actin (*CAG*) promoter. EGFP, *eGFP* gene. Library, oligo sequences containing the mutations/ motifs. RT, sequences for gene-specific reverse transcription. polyA, polyA signals. See also Supplementary Fig. 2. **b** Computational workflow. **c** Design of oligos containing rare variants from gnomAD and quantification of variant effects. $C_{DNA (ref)}$ represents DNA counts for the reference allele, $C_{RNA (ref)}$ represents RNA counts for the reference allele, $C_{DNA (alt)}$ represents DNA counts for the alternative allele, $C_{RNA (alt)}$ represents RNA counts for the alternative allele, $A_{ref}$ represents the activity score of the reference allele, $A_{alt}$ represents the activity score of the alternative allele, and lnFC represents the relative activity score of the alternative allele compared to the reference allele. Created with BioRender.com. **d** Diagram of oligos with random mutations in the motif and its flanking regions (22–23 nt). **e** Mismatch rate (%) per position along the length of DNA sequences harboring known motifs: SAMD4A, sterile alpha motif domain containing 4A motif (in gene *CHRDL1*), ARE, AU-rich element (in gene *CXCL2*), hPUM, human pumilio motif (in gene *MYOD1*), CDE, constitutive decay element (in gene *RBBP5*), and dPUM, *Drosophila* pumilio motif (in gene *SIPA1L2*). **f** Relative activity scores of each mutated position of known motifs in HEK293 (red) and HeLa (blue) cells. Relative activity scores are averaged across the 3 tested alternative alleles per position. The 95% confidence intervals are shown as shades. **e**, **f** Source data are provided as a Source Data file.

(±10 nt) of the tested variant was also low (Supplementary Fig. 5). The activities of biological replicates were highly correlated, signifying MapUTR's ability to reproducibly capture the impact of genetic variants on mRNA abundance (Fig. 2b). We also evaluated the statistical difference of the functional impacts of the two alleles via MPRAnalyze[42]. To call functional variants, we required a false discovery rate (FDR) ≤10% and a minimum absolute relative activity score of 10%. The latter cutoff was determined by inspecting the relative activity scores of mutations in known functional motifs, which mostly exceeded 10% (Fig. 1f).

Among the 17,210 and 17,194 rare variants tested by MapUTR in HEK293 and HeLa cells, respectively, 3814 (22%) and 4694 (27%) altered mRNA abundance significantly (Fig. 2c, Supplementary Data 1), with 6598 (38%) being significant in at least one cell line. Overall, the functional variants are relatively uniformly distributed across the length of 3′ UTRs (Supplementary Fig. 6a). These functional rare variants were harbored in 3992 genes, representing 52.1% of all tested genes. Among these variants, 50.9% and 51.2% in HEK293 and HeLa, respectively, had higher expression associated with the variant allele, whereas the rest of the variants downregulated mRNA expression. Moreover, among the 1,044 clinically relevant variants, 41.5% are functional in at least one cell line (26.2% in HEK293 and 26.7% in HeLa) (Fig. 2d). Importantly, 45% of these functional, clinically relevant variants are annotated in the ClinVar database with "uncertain significance". Thus, the MapUTR results shed light on the potential functionality of these variants.

To compare the effect sizes of the functional rare variants to those of common variants, we conducted another MapUTR experiment to test 3367 common 3′ UTR variants in HeLa cells (containing predicted functional variants in our unpublished work). A total of 1200 (35.6%) of these common variants were detected as functional by MapUTR. We observed that rare variants had significantly higher effect sizes than common variants (Supplementary Fig. 6b). This is consistent with the expectation that functional rare variants are generally more disruptive than common ones.

We next compared the function of 3′ UTR variants between the two cell lines. To this end, we correlated the relative activity scores of the 1910 (29% of the 6598) variants that were functional in both cell lines. A significant correlation was observed, and the majority of shared variants (~94%) showed the same direction of change between the two cell lines (Fig. 2e). The relative activity scores were also significantly correlated when all variants tested in both cell lines were included (Supplementary Fig. 6c). Thus, the genetic background, rather than *trans*-acting factors, plays a consistent and dominant role in determining the function of 3′ UTR functional variants between HEK293 and HeLa cells.

### Functional MapUTR variants alter miRNA target sites

The 3′ UTR is known to harbor *cis*-regulatory elements that recruit *trans*-acting factors, usually miRNAs and RBPs, for post-transcriptional regulation of gene expression[43]. To investigate the potential mechanisms associated with the rare functional MapUTR variants, we first

asked whether they may alter miRNA target sites[44]. In HEK293 and HeLa cells, 63.8% and 63.5% of all functional MapUTR variants overlapped predicted miRNA target sites, respectively (Methods). It should be noted that these percentages are not higher than the percentage of all tested variants overlapping miRNA target sites, possibly due to challenges in the accurate prediction of miRNA targets. Nonetheless, a functional variant disrupting miRNA targeting is expected to lead to enhanced mRNA abundance. Consistent with this expectation, in both HEK293 and HeLa cells, we observed a significant bias toward the upregulation of mRNA abundance by the alternative alleles in MapUTR (Fig. 3a). Although this effect is small, it indicates that miRNA target disruption may be one functional mode of 3′ UTR variants. Indeed, hundreds of miRNAs (726 in HEK293 and 988 in HeLa cells) were disrupted by MapUTR functional variants in their targets more often than by nonfunctional variants. For example, Fig. 3b shows two miRNA-target pairs. One miRNA, miR-34b-3p, is predicted to target the *PLIN4* transcript that harbors a rare variant (*rs767768172*) in the miRNA seed match region. MapUTR revealed significantly higher mRNA abundance associated with the alternative allele, consistent with the expected derepression by the miRNA in the presence of the rare variant. Similarly, another rare variant (*rs145078776*) is predicted to disrupt the binding of miR-3180-5p to the *LDHD* transcripts (Fig. 3b). It should be noted that both genes have important disease relevance, with *PLIN4* implicated in skeletal muscle disease[45] and *LDHD* involved in clear cell renal carcinoma[46].

### Functional MapUTR variants alter RBP binding sites

Next, we investigated the role of RBPs in affecting mRNA abundance of genes harboring rare functional MapUTR variants. As a first step, we examined the overlap of functional variants with RBP binding peaks in the ENCODE eCLIP data[47] from HepG2 and K562 cells. We observed that 57.6% or 57.5% of functional variants in HEK293 (2,195) or HeLa (2,701), respectively, resided within the binding peaks of at least one RBP. In addition, the functional variants were located closer to the eCLIP peaks than the nonfunctional ones (Supplementary Fig. 7). Next, we conducted motif analyses using HOMER[48] to identify over-represented hexamers among sequences that upregulated or down-regulated mRNA expression (see Methods). For sequences that downregulated mRNA expression, we identified well-defined destabilizing motifs such as the AU-rich and GU-rich elements (Supplementary Fig. 8a, c). In contrast, CU-rich, CA-rich, and GA-rich elements, which are known stabilizing motifs[49], were enriched among sequences that upregulated mRNA expression (Supplementary Fig. 8b, d). Additionally, the strength of the motifs was significantly altered by the MapUTR variants relative to nonfunctional or random control variants (Supplementary Fig. 8e, Methods). These results support the validity of the MapUTR experiment.

Next, we associated the above motifs with RBPs using previously published RNA Bind-n-Seq (RBNS)[50] data where binding motifs of RBPs were characterized experimentally (Supplementary Fig. 9). We then evaluated whether the alternative alleles of each functional variant altered RBP binding using the DeepRiPe model[51] (Methods). On the

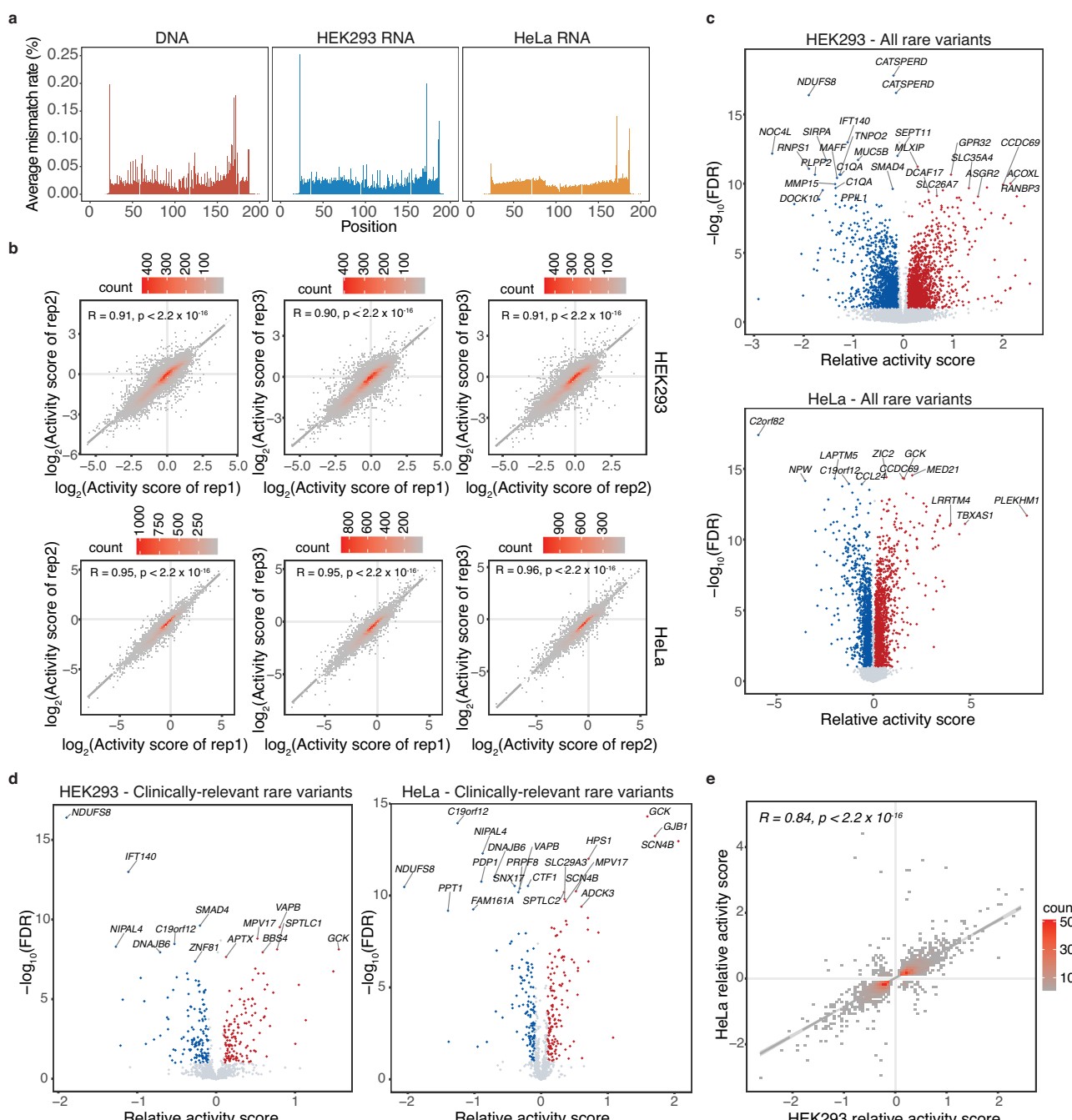

**Fig. 2 | MapUTR identifies functional rare 3′ UTR variants regulating mRNA abundance. a** Average mismatch rate (%) in DNA-seq and RNA-seq libraries. **b** Correlation of activity scores between biological replicates in HEK293 and HeLa respectively. *P*-value is based on a two-sided Spearman correlation test. **c** Relative activity scores and FDR of all rare 3′ UTR variants tested in HEK293 (top) and HeLa (bottom). **d** Relative activity scores and FDR of clinically relevant rare 3′ UTR variants tested in HEK293 (left) and HeLa (right). **e** Correlation of relative activity of functional variants common to HEK293 and HeLa cells. *P*-value is based on a two-sided Spearman correlation test. **a**–**e** Source data are provided as a Source Data file.

global level, we observed that functional MapUTR variants significantly altered RBP binding compared to nonfunctional or random dbSNPs (Fig. 3c, Methods). RBPs that are known to regulate RNA stability, such as HNRNPK, PCBP1 and ELAVL4, had significant binding score changes due to the functional variants, as predicted by DeepRiPe (Supplementary Fig. 10a). We further compared the relative activity of the variants measured by MapUTR and the predicted binding alteration by DeepRiPe. A number of RBPs showed significant correlations (Fig. 3d, Supplementary Fig. 10b). For instance, among functional variants, a negative correlation was observed for ZFP36, binding to the AREs WUUAUU and AUUUAU in HEK293 and HeLa, respectively (Fig. 3d),

consistent with the destabilizing function of AREs[52]. The nonfunctional variants, in contrast, showed virtually no correlation for the same RBPs and motifs. Altogether, these findings support the utility of MapUTR in identifying functional effects of variants that potentially alter RBP binding.

**Functional rare MapUTR variants enriched in genes with cancer-relevance**

To better understand the functional relevance of MapUTR variants, we first performed a Gene Ontology enrichment analysis of genes harboring the top 500 variants (ranked by their absolute relative activity in

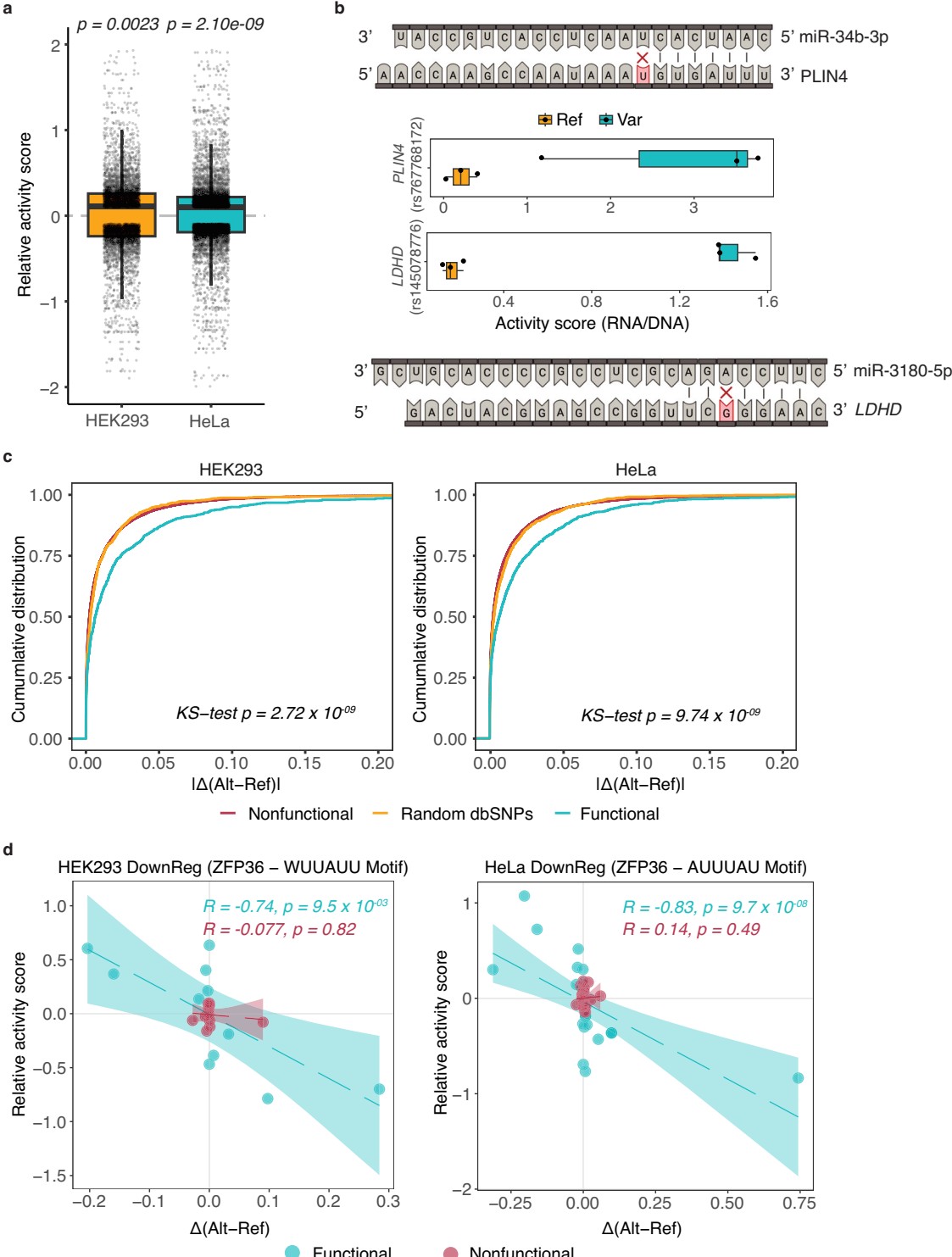

**Fig. 3 | Mechanisms of RNA abundance regulation via 3' UTR. a** Relative activity scores of functional variants that disrupt miRNA target sites. HEK293: $N = 6189$ unique variant-target pairs. HeLa: $N = 7569$ unique variant-target pairs. *P*-values are obtained by a one-tailed Wilcoxon rank sum test. **b** Functional variants in *PLIN4* (HEK293) and *LDHD* (HeLa) disrupt the target sites for miR-34b-3p (top) and miR-3180-5p (bottom), respectively. MapUTR activity scores are shown in the middle. $N = 3$ from biologically independent replicates. Created with BioRender.com. **c** Functional MapUTR variants significantly alter RBP binding (See also Supplementary Fig. 8, 9). X-axis shows the binding score difference between the reference and alternative alleles predicted by DeepRiPe. *P*-values were calculated using a two-tailed Kolmogorov-Smirnov test. The *p*-values indicated in the plot are the maximum *p*-values obtained from the two comparisons (functional versus

nonfunctional variants [HEK293: $p < 2.2 \times 10^{-16}$, HeLa: $p < 2.2 \times 10^{-16}$], and functional versus random dbSNPs [HEK293: $p = 2.72 \times 10^{-09}$, HeLa: $p = 9.74 \times 10^{-09}$]). **d** Relative activity of MapUTR variants is corroborated by predicted changes in RBP binding. X-axis similar as (**c**). Functional variants (blue) that increase ZFP36 binding to AU-rich elements WUUAUU and AUUUAU in HEK293 and HeLa, respectively, exhibit decreased gene expression. Nonfunctional variants (red) cause little changes in gene expression regardless of changes in RBP binding. The shaded bands indicate the 95% confidence interval from the line of best fit. See also Supplementary Fig. 10b. **a**, **b** All boxplots depict the median as the center line, the boxes define the interquartile range (IQR: 25th to 75th percentiles) and the whiskers extend up to 1.5 times the IQR. **a**–**d** Source data are provided as a Source Data file.

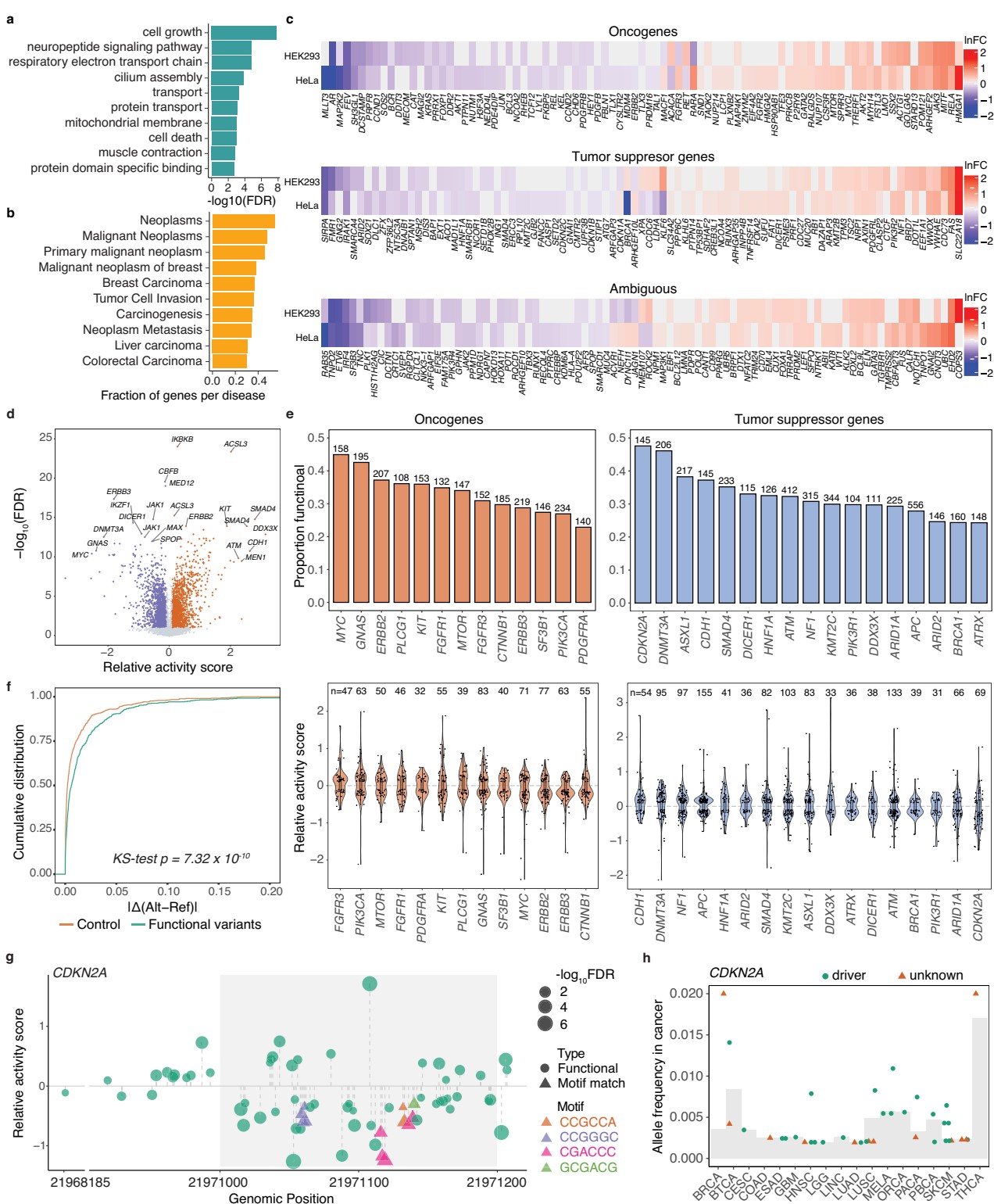

MapUTR of either HEK293 or HeLa cells). We observed that these genes were enriched in cell survival-related terms such as cell growth and cell death (Fig. 4a, Supplementary Fig. 11a, b). Next, we examined the association of these genes with different types of diseases via the DisGeNET database[53]. Interestingly, we found that cancer was the most represented disease (Fig. 4b), with 4,648 (70.4%) functional rare variants located in genes associated with cancer. Furthermore, 9 out of the 10 canonical oncogenic pathways[54] harbored genes with MapUTR functional variants, where genes in the PI3K/Akt, cell cycle, and Myc signaling pathways had

the highest overlap (Supplementary Fig. 11c). These results suggest that functional rare 3′ UTR variants likely play a role in tumorigenesis.

To further examine the relevance of functional rare variants to cancer, we asked whether cancer driver genes contain functional MapUTR variants. We compiled a list of cancer driver genes based on three different sources, including the Integrative OncoGenomics (IntOGen) databases[55], Pan-Cancer Analysis of Whole Genomes (PCAWG)[56], and the Catalogue of Somatic Mutations in Cancer (COSMIC) Cancer Gene Census tier 1 (v96)[40]. Among a total of 1,143 cancer

**Fig. 4 | MapUTR identifies functional variants in cancer driver genes. a** Gene ontology terms enriched in the genes with large-effect functional variants (top 500) in HEK293 and HeLa. Top 10 terms were plotted. **b** Disease associations most represented by MapUTR functional variants. Top 10 diseases were plotted. **c** Cancer driver genes containing MapUTR functional variants. For genes with multiple functional variants, the variant with the largest absolute relative activity was plotted in each cell line. **d** Relative activity scores and FDR of all cancer-related variants (COSMIC somatic mutations) tested in HeLa. **e** Top: Proportion of functional variants identified for cancer driver genes with at least 100 tested variants. The number on top of each bar denotes the number of variants tested for each gene. Bottom: Distribution of relative activity scores for oncogenes and tumor suppressor genes (TSGs) in the top panel. Boxplots depict the median as the center line, the boxes define the interquartile range (IQR: 25th to 75th percentiles) and the whiskers extend up to 1.5 times the IQR. **f** Changes in RBP binding caused by functional variants in cancer driver genes were significantly higher than expected. *P*-values were calculated using a two-sided Kolmogorov–Smirnov test. **g** Many functional variants in the 3′ UTR of the tumor suppressor gene *CDKN2A* decreased RNA abundance, indicating the potential for a cancerous outcome. Motif matches are denoted as triangles, with the specific motifs highlighted in different colors. Genomic positions (hg19) are shown on the x-axis. **h** Allele frequency (somatic) of down-regulating functional variants of *CDKN2A* in different cancers. The average allele frequency of all 3′ UTR somatic mutations in each cancer is indicated as a light gray bar. A variant is labeled as a driver if it was found in other driver mutation databases. **a**–**h** Source data are provided as a Source Data file.

driver genes, 267 had functional MapUTR variants (499 variants) (Fig. 4c, Supplementary Data 2). Importantly, 47 (54.0%) oncogenes harbored at least one functional variant leading to an increase of gene expression levels, whereas 55 (61.8%) tumor suppressor genes had at least one variant leading to a decrease of gene expression levels, potentially contributing to tumorigenesis (Fig. 4c). These results suggest that the functional rare MapUTR variants may be closely relevant to cancer.

## Functional 3′ UTR somatic mutations in cancer driver genes identified by MapUTR

Based on the above data on general rare variants, we hypothesized that mRNA abundance alteration may be a function common to many somatic mutations (mostly rare in the general population) in the 3′ UTR of cancer driver genes. Although many studies have aimed to identify functional mutations in cancer driver genes, the majority of these efforts has been centered on mutations occurring in the protein-coding regions[40,55–57]. Consequently, there remains a significant knowledge gap regarding the role of functional mutations in the 3′ UTRs that may contribute to cancer development via post-transcriptional regulation. To this end, we tested 11,929 COSMIC mutations[40] in the 3′ UTRs of 166 highly confident cancer driver genes[40,55,56] (Methods). Among these somatic mutations, 3,928 (33%) were confirmed to be functional by MapUTR (HeLa), affecting 155 genes (Fig. 4d, Supplementary Data 1). Note that a functional variant can be detrimental (pro-cancer), neutral or beneficial in tumors depending on its effect on mRNA abundance and the role of the cancer driver genes (tumor suppressing or oncogenic). It is possible that one detrimental mutation drive critical disease processes, despite the existence of many non-detrimental mutations. Interestingly, 56 of the 62 (90%) tumor suppressor genes had at least one variant that attenuates mRNA abundance, while 43 of the 50 (86%) oncogenes harbored at least one variant that enhances mRNA abundance, indicating that MapUTR can help to prioritize these detrimental variants.

For each cancer driver gene with more than 100 tested variants, a notable proportion (at least 23%) of their respective tested variants significantly altered mRNA abundance (Fig. 4e, top). In particular, tumor suppressor genes *CDKN2A* and *DNMT3A* had the highest proportions of functional variants (47.6% and 46.1%, respectively). All genes had functional variants with potentially detrimental effects (i.e., increased expression of oncogenes or decreased expression of tumor suppressor genes, Fig. 4e, bottom). Although each cancer driver gene may have many possible mutations, only a small number of mutations may occur in a particular patient and have a functional impact. Nonetheless, even a single functional mutation may cause detrimental effects by enhancing oncogene expression or repressing expression of a tumor suppressor.

Among the functional COSMIC variants in cancer driver genes, 405 were identified in at least one TCGA patient. We next examined the co-occurrence of these functional variants in each TCGA patient. We found that a small number of genes (e.g., *RET*, *RECQL4*, *CUX1*, *FGFR4*, *PIK3R1*, *FGFR3*, and *GNAS*) had some co-occurring variants per patient

in different cancer types (Supplementary Fig. 12a). Interestingly, variants co-occurred in the oncogenes *FGFR3*, *CCND1*, and *MYC* mostly increased mRNA abundance (Supplementary Fig. 12b). In contrast, those in the tumor suppressor gene *PIK3R1* and *ATM* mostly decreased mRNA abundance (Supplementary Fig. 12b). Thus, co-occurrence of these detrimental variants (pro-cancer) may exhibit synergistic effects on mRNA abundance regulation.

To understand how the functional MapUTR variants in cancer driver genes altered mRNA abundance, we conducted *trans*-factor analyses using a similar approach as for the gnomAD rare variants (Methods). Although miRNA target sites were not notably altered, our analysis showed that these variants significantly affected RBP binding. Specifically, sequences surrounding downregulating variants were enriched with the well-known destabilizing ARE (UAUUUA) motif among others (Supplementary Fig. 13a). Upregulating variants also showed significant enrichment with the CU-rich and GA-rich elements known to play stabilizing roles (Supplementary Fig. 13b). Once associating these motifs with RBNS data (Supplementary Fig. 13c, d) and analyzing the effects of variants on RBP binding via DeepRiPe, we observed significantly altered RBP binding strength by the functional variants (Fig. 4f, Supplementary Fig. 13e). These observations are consistent with the previous knowledge that RBP dysregulation impacts tumorigenesis or tumor progression[58], further suggesting that RBP dysregulation via 3′ UTR sequences is a potential mechanism implicated in cancer processes.

As an example, MapUTR revealed a total of 145 functional 3′ UTR variants in the tumor suppressor gene *CDKN2A*. This gene encodes two proteins p16[Ink4a] and p14[ARF], involved in the anticancer Rb and p53 pathways, respectively[59]. Thus, down-regulating functional variants in *CDKN2A* may weaken the Rb and p53 tumor suppressor pathways, potentially leading to cancer progression. The majority of the down-regulating functional variants in *CDKN2A* were enriched in a 200nt-long region (shaded in Fig. 4g). Through motif analysis, we identified several functional variants that created a destabilizing motif GCGACG (possible targets of RBM4 and RBM45 based on RBNS data, same below) or disrupted stabilizing motifs CCGCCA (possible targets of PCBP1, RBM6, HNRNPK, and SRSF5), CCGGGC (possible targets of RBM41, PRR3, and RBM22), or CGACCC (bound by PCBP1 and PCBP2) (Fig. 4g). Interestingly, the two motifs (CCGCCA and CGACCC) that can bind PCBP1 were very close to each other in the 3′ UTR of *CDKN2A* (Fig. 4g), suggesting a local region that might be sensitive to PCBP1-mediated regulation. Notably, a number of down-regulating functional variants in *CDKN2A* had elevated allele frequencies in multiple cancers (Fig. 4h), including skin cutaneous melanoma (labeled as SKCM or MELA) which is associated with *CDKN2A* mutations[60]. Besides *CDKN2A*, we also performed motif analysis for other tumor suppressor genes (Supplementary Fig. 14a–g) and oncogenes (Supplementary Fig. 14h, i). Indeed, many pro-cancer functional variants could be explained by RBP binding motifs. Altogether, MapUTR—coupled with the follow-up analysis on functional variants—can prioritize cancer-relevant variants and elucidate potential key RBPs that interact with the variant-containing sequences.

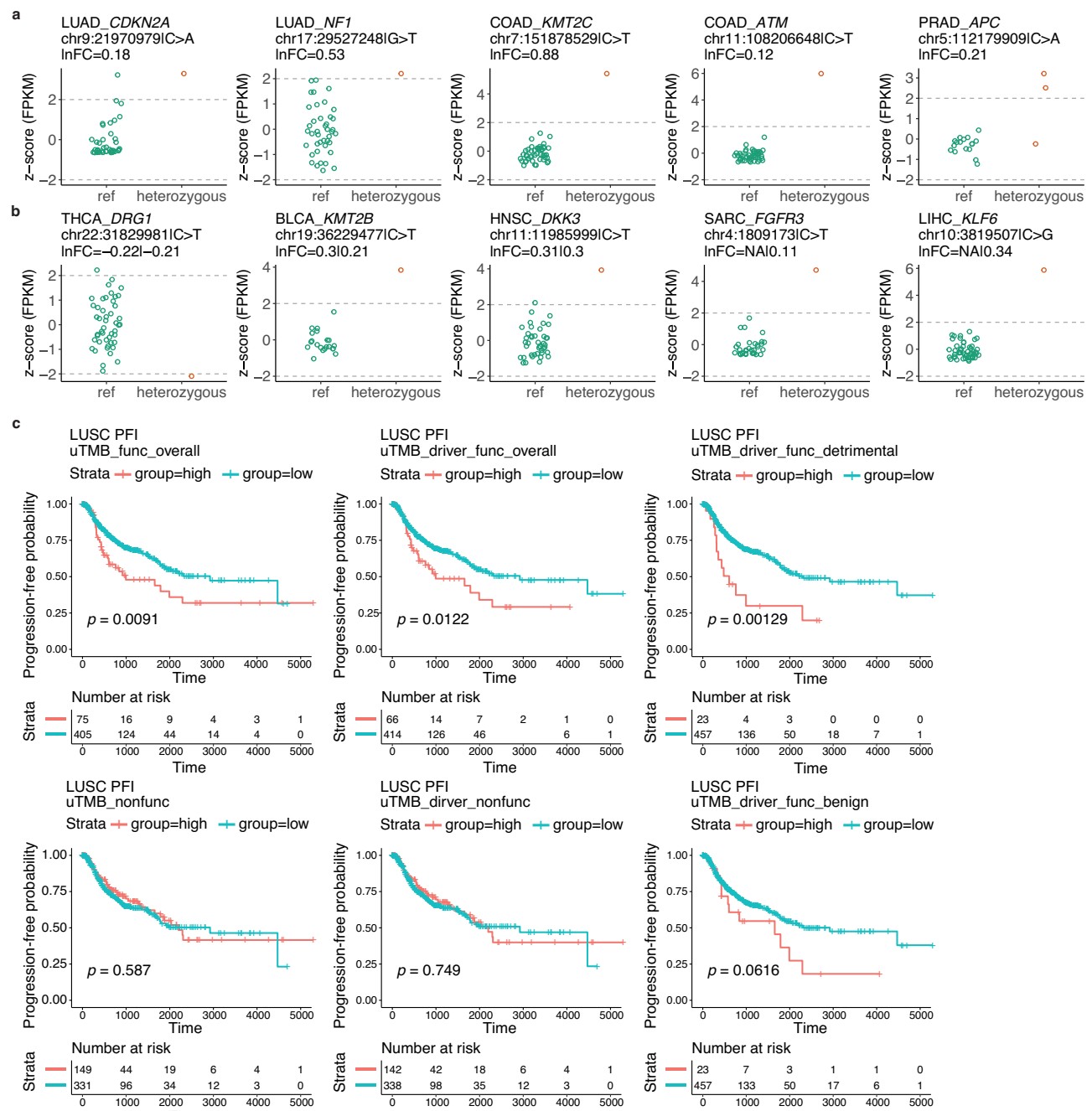

**Fig. 5 | Functional MapUTR variants are clinically relevant in cancer patients.**
**a** Functional COSMIC variants associated with gene expression outliers in different cancers. Cancer types and genes, genomic coordinates (hg19) and alternative alleles of the variant are shown on top. Relative activity scores (lnFC) tested in HeLa are also shown. **b** Similar as (**a**), but for functional rare variants. Relative activity scores (lnFC) tested in both HEK293 (before '|') and HeLa (after '|') are shown. **c** Higher uTMB of functional MapUTR variants is associated with worse progression-free interval (PFI) in LUSC. uTMB_func_overall: uTMB of functional variants in all tested genes, uTMB_driver_func_overall: uTMB of functional variants in all tested cancer driver genes, uTMB_driver_func_detrimental: uTMB of functional variants that increase oncogene expression or decrease TSG expression, uTMB_nonfunc: uTMB of non-functional variants in all tested genes, uTMB_driver_nonfunc: uTMB of non-functional variants in all tested cancer driver genes, uTMB_driver_func_benign: uTMB of functional variants that decrease oncogene expression or increase TSG expression. Patients were grouped into high (orange) and low (turquoise) groups by uTMB level tertiles. *P*-values were calculated by two-sided log-rank tests. **a–c** Source data are provided as a Source Data file.

## Functional MapUTR variants may underlie gene expression outliers in TCGA and GTEx

Next, we asked if the above functional COSMIC mutations in cancer driver genes may help to explain gene expression changes in cancer patients. We obtained genotype data including both germline and somatic mutations in The Cancer Genome Atlas (TCGA) from the Pan-Cancer Analysis of Whole Genomes (PCAWG)[56] database. For each

MapUTR-discovered functional COSMIC variant that was found in patients of a certain cancer type, we extracted gene Fragments Per Kilobase of transcript per Million mapped reads (FPKM) values in patients with either the reference or the alternative allele and calculated gene expression z-scores for outlier detection (Methods). A total of 113 functional variants were associated with patients whose gene expression value was an outlier (Fig. 5a, Supplementary Data 3). Given

our earlier observation of the enrichment of germline rare functional 3′ UTR variants in cancer-relevant genes, we expanded the outlier detection analysis to include all functional variants identified by MapUTR in this study. In total, we observed 508 functional variants (including both COSMIC and gnomAD variants) associated with gene expression outliers in multiple cancer types (Fig. 5b, Supplementary Data 3). We repeated the above analysis using GTEx data, which identified 352 functional variants with outlier expression and consistent direction of change as detected in MapUTR (Supplementary Fig. 15a). For each gene, only a few tissues showed outlier patterns (Supplementary Fig. 15b), indicating that there may exist heterogeneity in the variant effect as similarly noted by other studies[61,62]. These results indicate that the functional variants detected through MapUTR may help to interpret the mechanisms underlying gene expression outliers in cancer and normal tissues.

## uTMB calculated based on functional MapUTR variants predicts patient survival in LUSC

Given the roles of functional MapUTR variants in regulating cancer driver genes, we investigated their potential in predicting cancer patient survival. To this end, we defined a metric for each tumor sample, namely uTMB, to represent the number of somatic mutations that are functional MapUTR variants. Using the TCGA Pan-Cancer Atlas data[63,64], we observed that higher uTMB (uTMB_func_overall) was significantly associated with shorter progression-free interval (PFI) in lung squamous cell carcinoma (LUSC) (Fig. 5c). The observation also holds when only MapUTR variants in cancer driver genes (uTMB_driver_func_overall) or detrimental MapUTR variants (those that increased oncogene expression or decrease tumor suppressor gene expression, uTMB_driver_func_detrimental) were used (Fig. 5c). As controls, no significance was observed when the "uTMB" was calculated using variants confirmed as non-functional by MapUTR (uTMB_nonfunc or uTMB_driver_nonfunc), or variants that are benign (those that decrease oncogene expression or increase tumor suppressor gene expression, uTMB_driver_func_benign). Similar relationships were observed between uTMB and overall survival in head and neck squamous cell carcinoma (HNSC), albeit weaker than in LUSC (Supplementary Fig. 16). Overall, these findings highlight the potential clinical relevance of functional MapUTR variants, especially for LUSC patients.

## Functional rare MapUTR variants in *MFN2*, *FOSL2*, and *IRAK1* regulate mRNA stability and cell proliferation

MapUTR may be an effective means to nominate cancer driver mutations. To illustrate this capacity of MapUTR, we experimentally validated 3 functional MapUTR variants not previously known as cancer driver mutations. These three variants are located in three cancer-relevant genes. The first gene, Mitofusin 2 (*MFN2*), encodes a mitochondrial membrane protein regulating mitochondria fusion[65]. MFN2 has anti-tumor effects and is downregulated in multiple cancers[66]. Previous studies found that MFN2 inhibits cell proliferation by suppressing mTORC2/Akt or Ras-NF-κB signaling pathways[67,68]. In MapUTR, we identified a functional rare 3′ UTR variant (*rs777822288*) in *MFN2*, leading to a significant increase in mRNA expression (Fig. 6a). We hypothesized that this variant may also play a role in inhibiting cell proliferation. The second gene, FOS Like 2 (*FOSL2*), encodes a protein serving as a subunit of the transcription factor complex AP-1[69]. FOSL2 promotes cell proliferation, migration, and invasion in breast cancer and ovarian cancer[70,71]. We discovered one rare variant (*rs11884725*) in the 3′ UTR of *FOSL2*, which showed higher activity scores (RNA/DNA ratio) compared to the reference allele (Fig. 6a). This variant may facilitate cell proliferation by upregulating *FOSL2* gene expression. The third gene encodes interleukin-1 receptor-associated kinase 1 (IRAK1),

involved in the toll-like receptor and interleukin-1 signaling pathway[72]. Overexpressed in several cancers, IRAK1 is a therapeutic target, whose inhibition impairs tumor growth and metastasis[72]. We identified a rare 3′UTR variant (*rs782486025*) in *IRAK1* that significantly decreased mRNA expression (Fig. 6a). This variant may serve as an allele-specific 'inhibitor' for IRAK1, thus reducing cell proliferation.

To measure the effect of functional variants in their native genomic context, we utilized prime editing[73] to introduce the MapUTR variants into the genome of HEK293T cells, which have proven editing efficiency with prime editing[73] (Fig. 6b). In addition, HEK293T is a daughter cell line derived from HEK293 cell line, in which all three MapUTR variants were identified as functional candidates (Fig. 6a). Thus, HEK293T cells likely possess the *trans*-factors required for the MapUTR variant function. The genome-edited bulk HEK293T cells were diluted and plated to isolate single-cell clones for both the reference and variant alleles in each gene (Fig. 6b). To reduce potential bias due to off-target effects in a specific single-cell clone, we picked 4–6 single-cell homozygous clones for each allele (Fig. 6c–e). In addition, we checked the top three potential off-target sites predicted by CRISPRitz[74] via Sanger sequencing, for each epegRNA or nick sgRNA used in this study (Supplementary Table 3). Off-target editing was not observed in these sites (Supplementary Note 1).

Through quantitative reverse transcription PCR (qRT-PCR), we measured the mRNA stability of each gene in the single-cell clones by treating the cells with actinomycin D (ActD) to block cell transcription[75] for different time periods (2 h, 8 h, and 24 h) (Fig. 6f–h). Starting at 2 h post-ActD treatment, we observed significantly lower mRNA expression levels of *IRAK1* in the single-cell clones with *IRAK1* variant alleles compared to those with *IRAK1* reference alleles (Fig. 6h). This observation is consistent with the MapUTR result, in which the *IRAK1* variant allele had a lower activity score (RNA/DNA) compared to the reference allele (Fig. 6a). For *MFN2* and *FOSL2*, we observed a significant increase in mRNA expression levels in clones with the variant alleles at 8 h or 24 h post-ActD treatment (Fig. 6f, g), which are consistent with the higher MapUTR activity scores in the variant alleles for these two genes (Fig. 6a). These results confirm that the MapUTR variants in *MFN2*, *FOSL2*, and *IRAK1* regulate mRNA stability in HEK293T cells.

We next examined the functional impacts of the 3 MapUTR variants on cellular phenotype. To this end, we performed cell proliferation assays using the single-cell clones with either reference or variant alleles of *MFN2*, *FOSL2*, and *IRAK1* (Fig. 6b). We found that single-cell clones with the variant alleles of all three genes showed significantly altered cell proliferation profiles (Fig. 6i–k). Specifically, we observed reduced cell proliferation in clones with the variant alleles of *MFN2* (Fig. 6i) and *IRAK1* (Fig. 6k), as well as increased cell proliferation in clones with the variant allele of *FOSL2* (Fig. 6j). Importantly, the directions of the cell proliferation change are consistent with the expected consequence of each variant, based on their effects on mRNA stability and previous studies on the roles of *MFN2*, *FOSL2*, and *IRAK1* in cell proliferation[68,71,72].

To further explore the functional impacts of these three MapUTR variants, we conducted RNA-seq on the genome-edited single-cell clones. We identified 124, 37, and 2151 differentially expressed genes in the single-cell clones with alternative alleles of *MFN2*, *FOSL2*, and *IRAK1*, respectively (Supplementary Fig. 17a). For *MFN2*, the differentially expressed genes were enriched with those affecting the transforming growth factor-β (TGF-β) signaling pathway (Supplementary Fig. 17b), which regulates cell proliferation[76]. Similarly, the differentially expressed genes in single-cell clones with the *IRAK1* alternative allele were enriched in GO terms involved in cell proliferation regulation (Supplementary Fig. 17b). These observations are consistent with our findings above that these MapUTR variants altered cell proliferation profiles (Fig. 6i, k). Together, supporting the in vivo function of

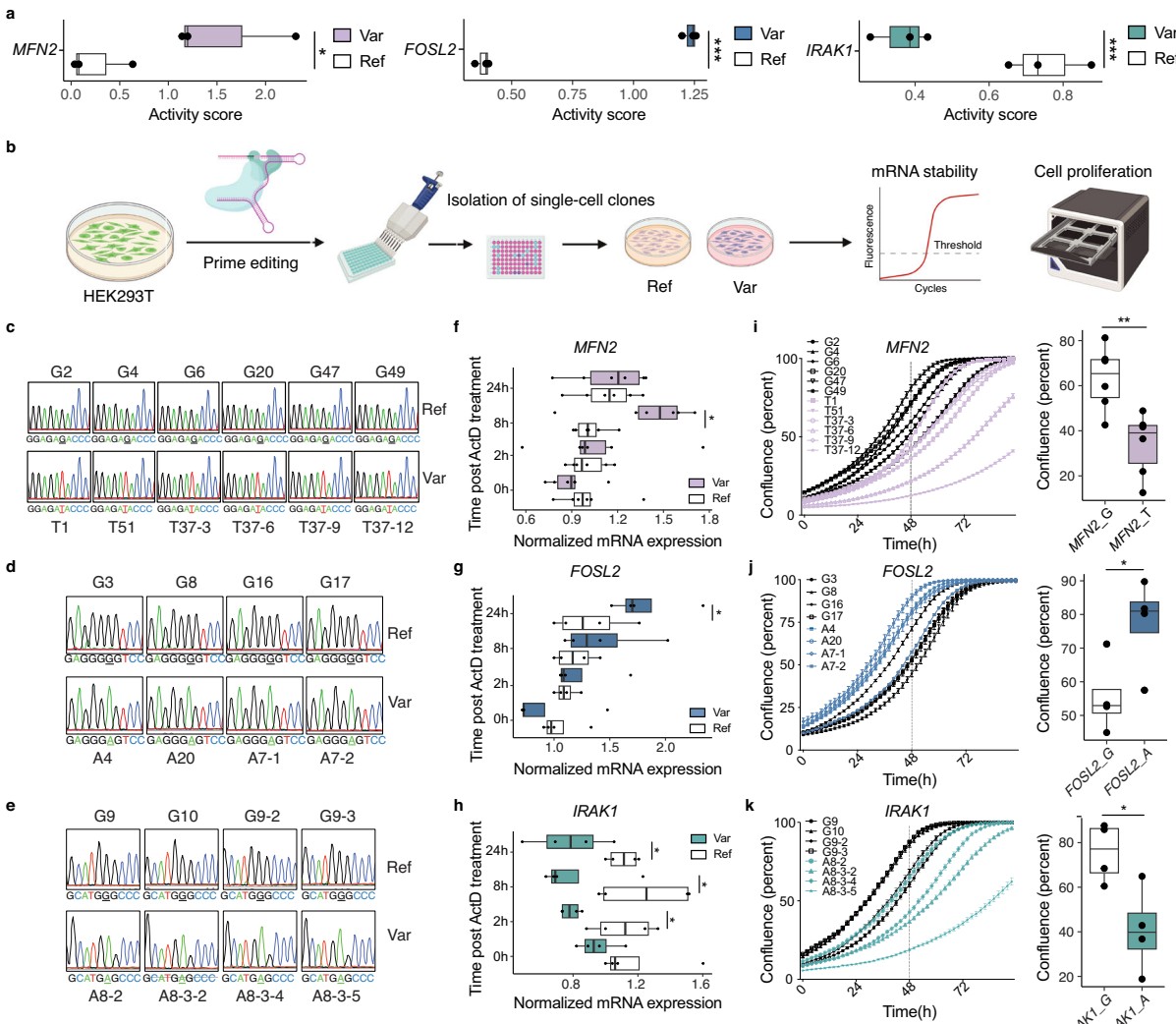

**Fig. 6 | Functional rare 3′ UTR variants regulate mRNA stability and cell proliferation in HEK293T cells. a** MapUTR activity scores of functional variants (*MFN2: rs777822288*, *FOSL2: rs11884725*, and *IRAK1: rs782486025*) measured in HEK293 cells. *P*-values were calculated using MPRAnalyze's two-sided likelihood ratio test and corrected using the Benjamini–Hochberg method (*N* = 3 biologically independent samples). *\*p* < 0.05, *\*\*p* < 0.01, *\*\*\*p* < 0.001. **b** Experimental validation workflow. HEK293T cells were transfected with plasmids expressing PEmax and epegRNAs to introduce the variant of interest. Single-cell clones were isolated and genotyped as either reference (Ref) or variant (Var) clones. The single clones were then used for downstream assays to test for mRNA expression/stability or cell proliferation. Created with BioRender.com. **c**–**e** Sanger sequencing results confirming the genotype of single clones with reference (Ref) and variant (Var) alleles for *MFN2* (**c**), *FOSL2* (**d**), and *IRAK1* (**e**). The alleles are underlined. **f**–**h** Normalized mRNA expression level of reference (Ref) and variant (Var) alleles of *MFN2* (**f**),

*FOSL2* (**g**) and *IRAK1* (**h**). Cells were treated with 10 μg/ml actinomycin D (ActD) and harvested at 2 h, 8 h, and 24 h post-treatment to test for mRNA stability. For *MFN2*, six biological replicates (*N* = 6 clones) for each allele were included in the experiment. For *FOSL2* and *IRAK1*, four biological replicates (*N* = 4 clones) per allele were included. *P*-values were calculated using one-tailed Student's *t* test. *\*p* < 0.05. **i**–**k** Cell proliferation assay of single clones with reference (Ref) and variant (Var) alleles for *MFN2* (**i**), *FOSL2* (**j**), and *IRAK1* (**k**). For *MFN2*, six biological replicates (*N* = 6 clones) for each allele were included in the experiment. For *FOSL2* and *IRAK1*, four biological replicates (*N* = 4 clones) per allele were included. The vertical dashed line indicates the cell confluence values at 48 h, which are plotted on the right. *P*-values were calculated using one-tailed Student's *t* test. *\*p* < 0.05, *\*\*p* < 0.01. **a**, **f**–**k** Boxplots are plotted as median, the 25% and 75% percentiles, and non-outlier maxima and minima. Specific *p*-values are provided in the Source Data file.

MapUTR variants in HEK293T cells, our data demonstrate the cancer-driving potential of these variants.

## Discussion

Rare variants constitute the majority of human genetic variants[1]. Yet, little is known about the function of non-coding rare variants due to their scarcity in the population, which has made rare variant association tests challenging[77]. In this study, we used a massively parallel reporter assay, MapUTR, to identify 3′ UTR variants regulating mRNA abundance post-transcriptionally. Based on the finding that a majority of functional gnomAD rare variants resided in cancer-relevant genes, we further tested the function of cancer somatic mutations via

MapUTR. Altogether, MapUTR uncovered 10,524 functional 3′ UTR variants (germline or somatic). This large catalog of functional variants enabled us to define a metric, uTMB, that quantifies TMB in untranslated regions. We showed the potential of uTMB in predicting patient survival in LUSC and HNSC, two types of squamous cell carcinoma. Our data reveal a trend of rare variants playing a role in affecting cancer-relevant genes and support the existence of many functional non-coding cancer mutations in 3′ UTRs, which facilitates the identification of non-coding cancer driver mutations.

It was previously reported that rare variants are enriched in gene expression outliers across tissues compared to non-outliers[78,79], indicating their pivotal role in regulating gene expression. Among all

tested gnomAD rare variants (17,301) in our experiments, 6,598 (38%) led to significant mRNA abundance alterations in at least one cell line, with an average of 24.5% variants identified as functional in each cell line. This prevalence of functional variants is much larger than previously reported for common 3′ UTR variants[18], where 19.5% (2,368 out of 12,173 total) were reported as functional variants in at least one of six human cell lines, with an average of 5.7% functional rate in each cell line. While the functional proportion may be arbitrary resulting from differences in the methods used to call functional variants in these two assays, a similar cutoff of an adjusted *p*-value less than 0.1 was applied in both assays, with MapUTR having an additional cutoff of relative activity ≥ 0.1. The higher proportion of functional variants in the rare variant screen may be explained by the purifying selection during evolution[80]. This observation is in line with a previous study that assessed the contribution of alleles with different allelic frequencies to gene expression in lymphoblastoid cell lines[81], revealing a higher contribution of rare variants to gene expression heritability.

We observed that the majority (70%) of functional rare 3′ UTR germline variants were in cancer-relevant genes, enriched in gene ontology terms such as cell growth and cell death (Fig. 4a, Supplementary Fig. 11a, b). However, the percentage of functional rare variants in cancer-related genes is not higher than that of all rare variants tested in MapUTR. Thus, it is interesting to note that, in general, gnomAD rare variants frequently occur in cancer-related genes, which is consistent with the expectation that rare variants are often detrimental and selected against, and the fact that we observed a larger effect size of functional rare variants, compared to common variants (Supplementary Fig. 6b). These data prompted us to analyze the other type of rare variants, somatic mutations, located in the 3′ UTRs of cancer driver genes, via MapUTR. Among the 11,929 COSMIC mutations we tested, 3,928 (33%) were detected to regulate mRNA abundance of their host genes in HeLa cells. This fraction is significantly larger than that of functional rare germline variants (4694 of 17,194 (27%)) detected in HeLa cells (*p* < 0.00001, proportion test). The above results suggest that somatic mutations located in 3′ UTRs have the potential to drive expression changes of important cancer driver genes. Indeed, altered gene expression, regulated by either DNA transcription or mRNA stability[82], is one of the major changes in cancer[83]. In a previous study on rare predisposition variants across 33 cancer types, around half of the variants located in tumor suppressor genes or oncogenes were associated with low or high gene expression, respectively[84]. Our data provide a genetic perspective that may account for the frequent gene expression changes observed in cancer.

Previous studies have primarily focused on identifying cancer driver mutations located in protein-coding regions, mainly due to limited methodologies and power in detecting non-coding cancer driver mutations. Indeed, non-coding driver mutations were believed to be less common than protein-coding drivers[85]. However, our data challenge this proposition by demonstrating the functional impact of over 10,000 rare germline or somatic mutations present in the 3′ UTRs of cancer-related genes. The large number of functional 3′ UTR variants reported in this study provides a valuable resource for future investigations into cancer driver mutations. Many of the genes that contain these functional mutations, such as *CDKN2A* highlighted in Fig. 5, are well-established cancer driver genes involved in multiple types of cancer. The functional implications of their 3′ UTR mutations should be tested thoroughly. As examples, we experimentally validated three rare variants in cancer-associated genes (*MFN2*, *FOSL2*, and *IRAK1*), confirming their functional roles in regulating mRNA stability and cell proliferation. Future studies characterizing additional functional 3′ UTR variants will help to elucidate their driver potential in cancer.

We introduce a concept, uTMB, that captures the somatic mutational burden in the non-coding genome of a tumor, specifically 3′UTRs. The calculation of uTMB is enabled by the large catalog of functional

MapUTR mutations. Traditionally, TMB is computed using somatic mutations in the coding regions, often focusing on non-synonymous mutations[86]. TMB has been extensively examined as a predictive biomarker of immunotherapy response[87]. Here, we showed that the metric uTMB calculated via functional MapUTR variants has the potential to serve as a predictive biomarker for LUSC and HNSC patient survival. Although it remains unknown why this relationship is most prominent in LUSC, our results support the possible clinical relevance of functional MapUTR variants. Since the function of genetic variants may be cell type- or tissue-specific, e.g., depending on RBP or miRNA expression, future studies should focus on specific cell types related to different types of cancer to uncover cancer-specific functional variants that improve the prognostic potential of uTMB. Our MapUTR assay is generally applicable to such studies, with a number of strengths, e.g., unbiased detection of both upregulating and downregulating variants (Fig. 2c), incorporation of UMI, and the minimal DNA/Cell ratio in cell transfection to mimic a physiologically relevant condition.

It is important to acknowledge that MapUTR comes with certain limitations. First, variants were tested in limited sequence context due to the oligo synthesis limit (up to 300 nt), preventing the identification of isoform-specific functional variants. Future endeavors, such as CRISPR editing-based screens combined with long-read sequencing, may be helpful in addressing this question. Additionally, only two cell lines (HeLa and HEK293) were tested with MapUTR. Both cell lines undergo active proliferation which may have contributed to the enrichment of cancer-relevant functional variants. Expanding the assay to more cell types would facilitate a better understanding of functional variants relevant to different biological processes in different tissues and diseases. Specifically, additional efforts are needed to address the low transfection efficiency in post-mitotic cells, e.g., using lentivirus to deliver the reporters. Third, in addition to functional variants affecting mRNA abundance, MapUTR may be modified to identify variants that affect alternative polyadenylation (i.e., creating an alternative proximal polyA signal), which could be a topic for future investigation.

The presence of outlier associations in TCGA or GTEx data provides support for the functional roles of a given variant. However, it should be noted that the absence of such associations does not necessarily imply non-functionality. Gene expression is intricately regulated by diverse mechanisms, and multiple genetic variants may collectively contribute to the regulation of a single gene. Consequently, the functional impact of a particular mutation may not manifest as a sole driver of gene expression (thus leading to outlier expression) but could be part of a broader network of regulatory elements.

In conclusion, our study uncovered more than 10,000 functional rare 3′ UTR variants regulating mRNA abundance post-transcriptionally, many of which reside in cancer driver genes. In general, the discoveries from MapUTR may help to nominate non-coding cancer driver mutations, uncover predictive biomarkers for patient survival, and explain heritability for complex diseases.

## Methods

### Generation of the MapUTR master plasmid

The MapUTR master plasmid was derived from the pEGFP-C1 plasmid (Clontech) with several modifications (See Supplementary Method, Supplementary Note 2: plasmid map). First, SacI restriction enzyme site was introduced via a synonymous G-to-C mutation in the *eGFP* gene to allow for easy cloning. Via this cloning site (NEB, Cat# R3156S) and a HpaI site (NEB, Cat# R0105S), extra sequences in the 3′ UTR of the *eGFP* gene were removed by re-cloning the *eGFP* and multiple cloning sites into the vector. Second, the CMV promoter was replaced with the CAG promoter, which was obtained from the CYP800 plasmid (gifted by Rockie Chong) via restriction enzymes NdeI (NEB, Cat# R0111S) and SacI-HF (NEB, Cat# R3156S). Third, the reverse transcription (RT) primer sequences, which were adapted from Illumina Read

1 sequencing primer sequences, were inserted between the multiple cloning site (MCS) and a polyA signal site by re-cloning the polyA signal fragment using BamHI-HF (NEB, Cat# R3136S) and MluI-HF (NEB, Cat# R3198S). Lastly, to avoid truncated transcripts mediated by alternative splicing at the end of the *eGFP* gene, we generated a synonymous G-to-A mutation (CAA**G**TAAGA to CAA**A**TAAGA) at the potential 5' splice site of the *eGFP* gene, which reduced the MaxEntScan score[88] from 7.61 to −0.57. Additionally, we also modified the end of the *eGFP* sequences that served as adaptors for library amplification. We introduced synonymous mutations (ACCTTA instead of ACTCTC) in the *eGFP* sequence such that the corresponding complementary primer had a lower chance of forming heterodimers with the P5 adaptors in other library amplifying primers. Primers used for master plasmid cloning are listed in Supplementary Data 4.

### Design of DNA oligonucleotides with random mutations within known motifs

Known 3' UTR *cis*-regulatory elements (Supplementary Table 1) were chosen from previous literature[25]. To test if MapUTR could capture the regulatory effects of these motifs, we designed oligos containing random mutations at every base (mutated to all 3 possible alleles respectively) within the regulatory motif region as well as its flanking regions (22–23 nt upstream and 22-23 nt downstream). The sequence context for the motif was chosen from an isoform that allows the motif to be at the center of the designed sequence. In case this was not possible, the sequence context was extended beyond the end of the annotated 3' UTR region. Each oligo in the oligo pool is 200 nt in length, with 158 nt being the tested 3' UTR sequences containing the variant of interest. The rest of the oligo contains forward primer binding site (21 nt), reverse primer binding site (15 nt), and restriction enzyme site EcoRI (6 nt) for cloning. All oligos were included in the oligo pool 1.

### Design of DNA oligonucleotides containing rare 3' UTR variants

We extracted human variants from the Exome Aggregation Consortium (ExAC)[89], now cataloged in the gnomAD[37], and excluded indels using GATK SelectVariants tool[90]. Further, with a threshold of adjusted minor allele frequency (MAF) less than 0.01, we obtained 1,017,886 rare variants. Based on GENCODE[91] basic v24 annotation, we selected 54,959 SNPs that were located in the 3' UTR. To avoid unwanted enzyme digestion and amplification within the oligos, we removed sequences that contain additional restriction enzyme sites and subpool primer sequences (Supplementary Table 2). We overlapped rare 3' UTR ExAC variants with a collection of clinically relevant variants reported in ClinVar[38], CIViC[39], COSMIC[40], and iGAP[41]. The resulting 1,044 SNPs, which we refer to as clinically relevant rare 3' UTR variants, were prioritized for final testing. In total, we included 17,301 variants, separated into three oligo pools (pool 1-3) for synthesis. Each oligo pool contains 3 to 4 subpools with different 5' and 3' adaptors that can be amplified using subpool primers, respectively (See Supplementary Method). Both reference and alternative alleles for each variant were included in the same subpool. In the oligo pool 2 and 3, each oligo is 200 nt in length, with 164 nt being the flanking sequence centered around the variant of interest. The sequence context for the motif was chosen from an isoform that allows the motif to be at the center of the designed sequence. In case this was not possible, the sequence context was extended beyond the end of the annotated 3' UTR region. The rest of the oligo contains forward subpool primer binding site (15 nt), reverse subpool primer binding site (15 nt), and restriction enzyme site EcoRI (6 nt) for cloning (Supplementary Table 2, design 1: F +rec1+lib+R). 2,737 rare variants were included in the oligo pool 1 with a similar oligo design but slightly shorter flanking sequences (158 nt instead of 164 nt), and one additional restriction enzyme site BamHI (Supplementary Table 2, design 2: F+rec1+lib+rec2+R).

### Cloning of synthesized oligonucleotides into MapUTR master plasmids

We resuspended the lyophilized oligo pools (Twist Bioscience) with ultrapure distilled water (Thermo Fisher Scientific, Cat# 10977015) at a final concentration of 1 ng/µl. Each subpool was amplified using subpool-specific primers (See Supplementary Method). The reverse subpool primer contains a BamHI restriction enzyme site, which allows subsequent digestion and ligation into the master plasmid. To avoid potential bias due to over-amplification, we first assembled qPCR reactions with PowerUp™ SYBR® Green Master Mix (Thermo Fisher Scientific, Cat# A25742) with 1 ng oligos as templates in a 50 µl reaction. We determined the cycle number with preliminary qPCR experiments to avoid overamplification of the oligo pools. With this cycle number (typically 17–19), we then repeated the PCR using the Q5 polymerase (NEB, Cat# M0492L). The PCR products were cleaned up via the DNA Clean & Concentrator kit (Zymo Research, Cat# D4004).

Next, DNA digestion reactions were set up for both the PCR products (oligo inserts) and master plasmids using the EcoRI-HF (NEB, Cat# R3101S) and BamHI-HF (NEB, Cat# R3136S) enzymes, followed by overnight incubation at 37°C. All digestion reactions were terminated with enzyme heat inactivation at 65°C for 20 min. For purification, the digested plasmids were resolved in 1% agarose gel and the desired band was gel purified using Zymoclean™ Gel DNA Recovery Kit (Zymo Research, Cat# D4002). The digested PCR products were directly cleaned up using the DNA Clean & Concentrator kit (Zymo Research, Cat# D4004). Cleaned-up PCR products and digested plasmids were ligated at a 10:1 molar ratio with the T7 DNA ligase (NEB, Cat# M0318). Ligation reactions were incubated at 25°C on a thermal cycler for 1 h, followed by a clean-up using the DNA Clean & Concentrator kit (Zymo Research, Cat# D4004), with water elution.

Finally, the purified ligation products were electroporated into the 10-beta Electrocompetent *E. coli* cells (NEB, Cat# C3020K) using the Gene Pulser Xcell Electroporation Systems (NEB, Cat# 1652660) at 2.0 kV, 200 Omega, and 25 µF. The transformed *E. coli* were spread onto 150 mm selective plates at 37°C overnight. Colonies to achieve at least a 100X coverage of the oligo library (e.g., 0.2 M colonies for 2000 designed oligos) were harvested for plasmid isolation using the ZymoPURE II Plasmid Midiprep Kit (Zymo Research, Cat# D4200). For an initial quality check, the isolated plasmid library was sent for Sanger sequencing with a sequencing primer (See Supplementary Method) complementary to the polyA signal region shared by all plasmids.

### Cell culture and transfection

HEK293 cells were gifted by Prof. Jing Huang at UCLA. HEK293 cells were not authenticated. HeLa cells were obtained from ATCC (Cat# CCL-2). HEK293 and HeLa cells were maintained in DMEM (Gibco, Cat# 11995065) with 10% FBS (Gibco, Cat# 26140079) and antibiotic-antimycotic reagent (Gibco, Cat# 15240062) at 37°C with 5% $CO_2$ supply. Cells were passaged the day before electroporation to make sure they were actively dividing by the time of the electroporation. Prior to electroporation, HEK293/HeLa cells were disassociated with Trypsin-EDTA (Gibco, Cat# 25300120), washed with growth media, and resuspended in OptiMEM (Gibco, Cat# 31985062) at a cell density of 10 M/ml. For a typical subpool library with 0.2 million (M) colonies, 3 µg plasmid library was electroporated into 15 M HEK293/HeLa cells (See DNA/Cell ratio optimization below) for each biological replicate, for a total number of three biological replicates. Electroporation was carried out using the Gene Pulser Xcell Electroporation System (NEB, Cat# 1652660) with the following settings: square wave, 25 msec, 220 V, 0.4 cm. After electroporation, HEK293 and/or HeLa cells were incubated in growth media at 37°C for 24 h.

### mRNA isolation

Twenty-four hours after electroporation, HEK293/HeLa cells were lysed using TRIzol (Thermo Fisher Scientific, Cat# 15596026). Each 500

µl TRIzol-lysed solution was mixed with 100 µl chloroform (Fisher Chemical, Cat# C298-500) to allow phase separation. The upper aqueous phase was transferred and mixed with equal volume ethanol (200 proof, Fisher BioReagents). The mixture was loaded to the column supplied by the Direct-zol RNA Miniprep Plus kit (Zymo Research, Cat# R2072) to isolate total RNA following the manufacturer's protocol. PolyA selection was carried out to isolate mRNA from total RNA using Dynabeads™ Oligo(dT)$_{25}$ (Thermo Fisher Scientific, Cat# 61002). The concentration of mRNA in each sample was quantified using the Qubit RNA HS Assay Kit (Thermo Fisher Scientific, Cat# Q32852) with the Qubit 2.0 Fluorometer (Thermo Fisher Scientific).

### Generation of UMI-containing libraries

We generated UMI-containing libraries from the plasmid library (DNA) before electroporation, as well as mRNA isolated after electroporation. To compare the DNA libraries made from plasmids before and after electroporation, we also isolated plasmid DNA from the transfected cells using PureLink™ Quick Plasmid Miniprep Kit (Thermo Fisher Scientific, Cat# K210011).

The mRNA was reverse transcribed into cDNA using the Super-Script™ IV Reverse Transcriptase (Thermo Fisher Scientific, Cat# 18090010) and a gene-specific reverse transcription (RT) primer (MPP3) that contains a 15-mer unique molecular identifier (UMI), which was synthesized as -NNNNNNNNNNNNNNN- (See Supplementary Method for primer sequences). After RT, mRNA was removed via RNase H treatment.

Both cDNA and plasmid DNA underwent two rounds of PCR (Supplementary Fig. 1). The first-round (3 cycles) utilized primers (MPP2 & MPP3) to add UMIs to the cDNA or plasmid DNA samples. The low-cycle number was utilized to minimize PCR amplification bias. The first-round PCR products were cleaned up using the DNA Clean & Concentrator kit (Zymo Research, Cat# D4004). Next, a second-round PCR was performed using the purified first-round PCR products and primers (MPP2 & MPP4), which added sample indexes and Illumina sequencing adaptors (P5/7). To avoid over-amplification, a pilot reaction for the second-step PCR was performed using the PowerUp™ SYBR® Green Master Mix (Thermo Fisher Scientific, Cat# A25742) and ran on a qPCR thermal cycler. An amplification curve was obtained for each sample to determine the cycle number before the plateau. This cycle number (or a smaller value) was then used in the second-round PCR (See Supplementary Method). All PCR steps for sequencing library generation were performed using the Q5 polymerase (NEB, Cat# M0492L).

PCR reactions for the same sample were pooled and purified using the DNA Clean & Concentrator kit (Zymo Research, Cat# D4004). Purified PCR products were resolved on a 2% agarose gel and the band at the expected library size (377 bp) was cut out and purified using the Zymoclean™ Gel DNA Recovery Kit (Zymo Research, Cat# D4002). UMI-containing libraries made from DNA/RNA were mixed and sequenced using custom sequencing primers (See Supplementary Method) on Illumina Hiseq3000 PE150 or Novaseq SP PE150 with 15% PhiX spike-in.

### DNA/Cell ratio transfection optimization

To optimize the DNA/Cell ratio for MapUTR, plasmid libraries were electroporated into 7.5 M HEK293 cells with increasing amount of DNA/Cell ratios (i.e., 40 ng/1 M, 200 ng/1 M, and 4 µg/1 M), respectively. Total RNA was isolated 24 h following electroporation. UMI-containing libraries were generated from the plasmid libraries before electroporation and the mRNA isolated from transfected cells (see details above). To verify the allelic ratios for the three control SNPs in the genes CXCL2[25] and ESR1[33], each control gene was amplified by a gene-specific reverse primer and the common P5 forward primer (See Supplementary Data 4). To avoid overamplification of the target genes, a qPCR reaction was assembled for each condition to determine the

cycle number before the plateau. Control genes were then amplified from each UMI library with the cycle numbers determined by qPCR using the Q5 polymerase (NEB, Cat# M0492L). PCR amplicons were gel-purified and sent for Sanger sequencing. The allelic ratio for each SNP was estimated based on the peak signal for each base, which was quantified using 4Peaks (Nucleobytes). RNA/DNA ratios were calculated by dividing the allelic ratios in the RNA samples by the allelic ratios in the DNA samples.

### Mismatch rate analysis for DNA and RNA reads

To assess the quality of sequencing data obtained from MapUTR, we examined the mismatch rate at any given position of the designed sequences. For each position that was covered by sequencing reads, we calculated the mismatch rate as follows:

$$Mismatch\ rate\ (i) = \frac{Number\ of\ reads\ with\ mismatches\ at\ position\ i}{Total\ number\ of\ reads\ covering\ position\ i} \times 100 \tag{1}$$

The 15 nt primer sequence on the ends of the designed sequence was excluded from this calculation.

### Estimation of variant effect sizes

Paired-end reads (2 x 150 nt) were obtained for each DNA and RNA library (3 biological replicates of each type). Read 1 contains a UMI (15 nt), reverse transcription (RT) primer (14 nt), REC2 restriction enzyme site (6 nt), subpool primer (15 nt), and 100 nt of the designed sequence. Read 2 consists entirely of the designed sequence. UMIs, together with the RT primer, REC2 restriction enzyme site, and subpool primer, were extracted from read 1 and added to the read name using UMItools[92]. The reads were then aligned to the reference sequences using Bowtie 2[93], allowing up to 1 mismatch per alignment. We only retained perfectly mapped reads or reads with 1 mismatch at positions other than the designed SNPs.

PCR duplicates were removed by retaining only one read with the same UMI that mapped to the same reference sequence. The UMIs were then counted in the DNA- and RNA-seq libraries, respectively. The counts were quantile-normalized across the 3 biological replicates. For each allele of a SNP, we calculated its activity score as follows:

$$Activity\ Score\ (A) = \frac{C_{RNA}}{C_{DNA}}, \tag{2}$$

where $C_{RNA}$ is the normalized RNA count and $C_{DNA}$ is the normalized DNA count for the allele.

The relative activity score of the two alleles of a SNP was defined as:

$$Relative\ Activity\ Score\ (\ln FC) = \ln\left(\frac{A_{alt}}{A_{ref}}\right), \tag{3}$$

where $A_{alt}$ is the activity score of the alternative allele and $A_{ref}$ is the activity score of the reference allele.

To compare the relative impact of the alternative alleles of a variant on mRNA abundance, we modeled RNA counts as a function of DNA counts using MPRAnalyze[42]. To call functional variants, we required an FDR ≤ 10% and a minimum absolute relative activity score of 10%. The latter cutoff was determined by inspecting the relative activity scores of mutations in known functional motifs, which mostly exceeded 10% (Fig. 1f).

### Analysis of MapUTR variants in miRNA target sites

We obtained genomic locations of all TargestScan miRNA target site predictions (version 7.2)[94], including both conserved and non-conserved families and sites. We required each predicted miRNA

target site to have a minimum context++ score percentile of 50 as recommended by the authors of TargetScan. We then examined the overlap between functional MapUTR SNPs (reference alleles) and predicted miRNA seeds, thus focusing on variants whose alternative alleles disrupted miRNA targets. Next, we used a one-tailed Wilcoxon rank sum test to evaluate whether the overlapping variants had relative activity scores biased toward upregulation.

## Motif discovery

To conduct motif analysis, we first grouped the reference and alternative sequences based on their observed effects in the MapUTR results. Briefly, we compared the mRNA abundance of the reference and alternative allele of each rare variant. If the alternative allele yielded higher expression than the reference, the sequence containing the alternative allele was included in the upregulating sequence group, and the reference allele-harboring sequence in the downregulating sequence group, and vice versa. Subsequently, we obtained a 11 nt sequence around the variant position (5 nt on each side) for the reference and alternative alleles, respectively. For the upregulating and downregulating sequence groups, respectively, a de novo motif search of the RNA sequences was conducted with HOMER[48], with an upper limit of 25 motifs. Note that the downregulating sequences were used as background when identifying motifs in the upregulating set, and vice versa.

## Motif strength analysis

To evaluate whether a variant altered the strength of the motifs more significantly than expected, we carried out the following motif strength analysis. Given the position-weight matrix (PWM) of a motif and assuming a uniform background distribution of bases, we scored all sequences containing the motif for changes in strength upon the presence of the variant. As a control, we shuffled the PWM and calculated the change in motif strength again. Then, we compared the distribution of changes in the true motif to the control for significant differences using a two-sided Kolmogorov–Smirnov test. We also incorporated another set of controls by randomly sampling the same number of nonfunctional variants as the functional ones and scoring them against the motifs. The change in motif strength was calculated as follows:

$$\Delta_{\mathrm{var}} = |S_{\mathrm{var}} - S_{ref}| \qquad (4)$$

where the strength of the motif with the variant allele, $S_{\mathrm{var}}$, is defined as $S_{\mathrm{var}} = \frac{\Pi p_i}{0.25^n}$. Similarly, the strength of the motif with the reference allele, $S_{ref}$, is defined as $S_{ref} = \frac{\Pi p_i}{0.25^n}$, where $p_i$ is the probability of the base at position $i$ of the PWM and $n$ is the length of the motif.

## Integrative analyses of RBPs and discovered motifs

To examine whether functional variants from MapUTR were located in RBP binding sites, HOMER-identified motifs were matched with motifs of each RBP reported by a previous RBNS study[50]. For each variant, we used the HOMER-identified motif and its associated RBPs from RBNS to assess the effect of the variant on RBP binding. Specifically, we used the DeepRipe model[51] to calculate the predicted difference in RBP binding between the reference and alternative alleles of a variant. As controls, nonfunctional variants and random rare dbSNPs were separately sampled matching the number of tested variants per chromosome. Subsequently, for each RBP, control SNPs were randomly chosen to match the number of functional variants bound by the RBP. Then, the control SNPs were scored with DeepRiPe similarly as for functional variants. The distribution of absolute changes in binding (reference versus alternative alleles) was compared between the functional variants and random control SNPs. These steps were conducted for data derived from HEK293 and HeLa cells respectively.

## Gene ontology (GO) enrichment analysis

Functional variants from MapUTR experiments in HEK293 and HeLa cells were combined to generate the top 500 unique variants ranked by their relative activity. The resulting 324 genes were chosen as the query genes for GO enrichment analysis. GO analysis was also conducted for each cell line separately using a similar strategy (436 and 404 query genes in HEK293 and HeLa, respectively). For each query gene, a control gene was randomly chosen from the background genes (excluding all query genes) tested in MapUTR. In this way, a control set of genes was constructed that has the same number of genes as the query set. This process was repeated 10,000 times. To evaluate the enrichment of each GO term in the query genes, a $p$-value was calculated by fitting a normal distribution to the occurrence frequencies of the same GO term in the 10,000 sets of controls. A FDR < 0.05 and occurrence (number of genes associated with a term) $\geq 5$ were required for significance. GO terms were ranked based on $-\log_{10}(FDR)$, and the top 10-16 GO terms were plotted.

## Cancer driver genes in MapUTR results for rare variants

A list of cancer driver genes was compiled from three different sources: (1) Integrative OncoGenomics (IntOGen) databases[55] (https://www.intogen.org/download), (2) Pan-Cancer Analysis of Whole Genomes (PCAWG)[56], (3) Catalogue of Somatic Mutations in Cancer (COSMIC) Cancer Gene Census tier 1 (v96)[40]. All cancer driver genes were divided into three groups (oncogenes, tumor suppressor genes, or ambiguous). Genes were labeled as ambiguous unless an agreement could be achieved across the three sources. Genes containing MapUTR functional rare variants were overlapped with the cancer driver genes. To plot the heatmap of the overlapped cancer driver genes, the variant with the largest relative activity (absolute value) was reported for each gene in each cell line.

## Design of DNA oligos containing 3′ UTR somatic mutations in cancer driver genes

A subset of highly confident cancer driver genes was defined by taking an intersect of genes listed in the IntOGen databases[55], PCAWG studies[56], and COSMIC Cancer Gene Census with the hallmark descriptions (v96)[40]. From GENCODE[91] comprehensive v41 annotation, we extracted the 3′ UTR regions of the highly confident cancer driver genes (180 genes) and overlapped them with all the non-coding somatic mutations cataloged in COSMIC (v96). We applied a sample count (CNT) filter of greater than 1 to make sure one mutation is supported by at least two samples, resulting in 14,579 mutations to be tested with MapUTR. We further removed variants whose flanking sequences contain restriction enzyme sites or subpool primer sequences that rendered them incompatible with the MapUTR cloning steps (Supplementary Table 2). Following these filters, we included 11,929 somatic mutations in 166 highly confident cancer driver genes in the oligo pool 4. Each oligo in the oligo pool is 200 nt in length, including forward subpool primer binding site (15 nt), restriction enzyme site EcoRI (6 nt), flanking sequence centered around the variant of interest (164 nt), and reverse subpool primer binding site (15 nt).

## Functional testing of 3′ UTR somatic mutations in cancer driver genes via MapUTR

The same MapUTR procedures as described for rare variants were applied to test the somatic mutations in cancer driver genes. Briefly, oligos from the oligo pool 4 were cloned into the master plasmid via restriction enzymes EcoRI-HF (NEB, Cat# R3101S) and BamHI-HF (NEB, Cat# R3136S). The plasmid libraries were then electroporated into HeLa cells (at 200 ng/1 M DNA/Cell ratio), which were lysed after incubation in 37°C with 5% $CO_2$ supply for 24 h. The plasmid libraries and the isolated mRNA were used to make UMI-containing libraries

(See Supplementary Method). The libraries were pooled and sequenced on Novaseq SP&S1 PE150 with 15% PhiX spike-in.

## Allele frequency of functional somatic mutations in cancer

Aggregated simple somatic mutation data was downloaded from the ICGC Data Portal (https://dcc.icgc.org/releases/release_28/Summary). For a variant that was found in multiple projects for the same type of cancer, the allele frequency of this variant in each project was plotted as an individual data point under the same cancer type (Fig. 4h). The background level was defined as the average allele frequency of all 3′ UTR single nucleotide variants reported in the aggregated file for each cancer type. The 3′ UTR somatic mutations were extracted by overlapping the aggregated list of mutations with GENCODE[91] comprehensive v41 annotation.

## Expression outlier detection in TCGA and GTEx

Genotype data including both germline and somatic mutations in The Cancer Genome Atlas (TCGA) was obtained from the Pan-Cancer Analysis of Whole Genomes (PCAWG)[56] through the ICGC Data Portal (http://dcc.icgc.org/pcawg/). FPKM data was downloaded from the Genomic Data Commons (GDC) portal (https://portal.gdc.cancer.gov/). For MapUTR variants that are present in TCGA samples, z-scores of gene FPKM values were calculated for individuals with reference alleles (homozygous) or variant alleles (heterozygous or homozygous variant). An expression outlier is defined by a z-score > 2 or z-score < −2. For GTEx data, genotype and gene expression data were obtained for 838 individuals across 55 tissues from the GTEx portal (https://www.gtexportal.org/). Outlier expression was detected similarly as for TCGA data. Using Fisher's exact test, we compared the proportion of samples with functional variants that are outliers to the proportion of homozygous reference samples that are outliers in each gene and in each tissue.

## Quantification of untranslated tumor mutational burden (uTMB) in TCGA

We obtained somatic mutations of 33 tumor types from the TCGA Pan-Cancer Altas[63] (http://api.gdc.cancer.gov/data/1c8cfe5f-e52d-41ba-94da-f15ea1337efc). For each patient, we extracted the somatic mutations from the aggregated file and overlapped them with the functional MapUTR variants. We then counted the total number of somatic mutations that are also functional MapUTR variants as the untranslated tumor mutational burden (uTMB_func_overall) for that patient. We also calculated uTMB using only the functional MapUTR variants in cancer driver genes (uTMB_driver_func_overall). Depending on the relative activity score of each variant in cancer driver genes, we further defined detrimental variants (uTMB_driver_func_detrimental) as those that increase oncogene expression or decrease tumor suppressor gene expression, and benign variants as those decreasing oncogene expression or increasing tumor suppressor gene expression were considered benign (uTMB_driver_func_benign). For control purposes, we also calculated the uTMB using non-functional MapUTR variants in all tested genes and cancer driver genes (uTMB_nonfunc and uTMB_driver_nonfunc).

## Survival associations

To obtain progression-free interval and survival times in cancer patients, we acquired the TCGA Pan-Cancer Clinical Data Resource[64] from https://api.gdc.cancer.gov/data/1b5f413e-a8d1-4d10-92eb-7c4ae739ed81. Patients were divided into high and low uTMB groups based on tertiles of uTMB levels in each cancer type. We compared progression-free interval and overall survival between the two groups using two-sided log-rank tests in different cancer types with the R package survival. Significance was determined with a p-value of < 0.05. We then generated the Kaplan-Meier survival curves using the R package survminer.

## Generation of single-cell clones containing MapUTR variants via prime editing

To introduce a MapUTR variant to the HEK293T genome using prime editing[73,95,96], the spacer and extension sequences for epegRNAs and nick gRNAs were designed using pegFinder[97]. A linker pattern was designed for each epegRNA using pegLIT[96]. For epegRNA constructs, the spacer, extension (contains a unique linker), and pegRNA scaffold sequences (See Supplementary Data 4) were cloned into the pU6-tevopreq1-GG-acceptor vector (Addgene, Plasmid #174038) via Golden Gate assembly. Similarly, the spacer and nick sgRNA scaffold sequences (See Supplementary Data 4) were cloned into the pU6-pegRNA-GG-acceptor vector (Addgene, Plasmid #132777) to generate nick gRNA-expressing constructs.

HEK293T cells were obtained from ATCC (Cat# CRL-11268). HEK293T cells were not authenticated. HEK293T cells were maintained in DMEM (Gibco, Cat# 11995065) with 10% FBS (Gibco, Cat# 26140079) and the antibiotic-antimycotic reagent (Gibco, Cat# 15240062) at 37°C with 5% $CO_2$. HEK293T cells were seeded in 48-well plates to reach 50% confluency by the time of transfection. The enhanced prime editing system[73], consisting of plasmids expressing the epegRNA (250 ng), nick gRNA (83 ng), and prime editor (750 ng), i.e., pCMV-PEmax-P2A-hMLH1dn (Addgene, Plasmid #174828), was used for cell transfection. For MFN2, only the epegRNA and prime editor were used for cell transfection due to a higher editing efficiency compared to the other strategy that includes an additional nick gRNA. Cell transfection was performed with the Lipofectamine™ 3000 Transfection Reagent (Thermo Fisher Scientific, Cat# L3000015) according to the manufacturer's protocol. For genotyping, genomic DNA (gDNA) was extracted and amplified with primers specific to each candidate variant (See Supplementary Data 4). PCR amplicons were purified and sent for Sanger sequencing with one of the PCR primers. Three days after cell transfection, the transfected cells were re-plated into 96-well plates by serial dilution to generate single-cell clones. Single-cell clones were then expanded and genotyped via Sanger sequencing.

To examine the potential off-target effects of prime editing, we tested three off-target sites predicted by CRISPRitz[74], for each epegRNA or nick sgRNA applied to the cells. Specifically, we searched for off-target sites allowing 4 mismatches, 1 DNA bulge, and 1 RNA bulge. We further ranked and selected the top 3 off-target sites based on the cutting frequency determination (CFD) score[98]. Next, we performed PCR and Sanger sequencing using the isolated genomic DNA of each single-cell clones to check for off-target editing.

## Measurement of mRNA expression levels via qRT-PCR

Single-cell clones with MapUTR variants were maintained in DMEM (Gibco, Cat# 11995065) with 10% FBS (Gibco, Cat# 26140079) and antibiotic-antimycotic reagent (Gibco, Cat# 15240062) at 37°C with 5% $CO_2$. For RNA isolation, cells were washed with PBS (Gibco, Cat# 14190144) and lysed with TRIzol (Thermo Fisher Scientific, Cat# 15596026). Total RNA was isolated using the Direct-zol RNA Miniprep Plus kit (Zymo Research, Cat# R2072) following the manufacturer's protocol. 1 to 2 μg of total RNA was used for cDNA synthesis with SuperScript™ IV Reverse Transcriptase (Thermo Fisher Scientific, Cat# 18090010) using random hexamers. To measure mRNA expression levels of genes containing MapUTR variants, 1 μl cDNA was used for qPCR reactions using the PowerUp™ SYBR® Green Master Mix (Thermo Fisher Scientific). Primers used for qPCR (same as gDNA PCR primers) are listed in Supplementary Data 4. The reaction was performed in the CFX96 Touch Real-Time PCR detection system (Bio-Rad) with the following settings: 50°C for 10 min, 95°C for 2 min, 95°C for 15 s, 60°C for 30 s, and with the last two steps repeated for 45 to 55 cycles. The expression of genes containing MapUTR variants (MFN2, FOSL2, and IRAK1) was normalized against the expression of TBP. For mRNA stability assays, single-cell clones with MapUTR variants were treated with 10 μg/ml actinomycin D (Sigma-Aldrich) in growth media.

Cells were harvested at different time points (2 h, 8 h, and 24 h) post-actinomycin D (ActD) treatment for RNA isolation and RT-qPCR. Two technical replicates were performed for each single-cell clone during RT-qPCR. For each gene, 4 to 6 single-cell clones were used for either reference or variant alleles. Samples of reference and variant alleles collected at the same time point were analyzed in one PCR plate to allow for proper comparisons. *P*-values were calculated using one-tailed Student's *t* test. To call significance, *p*-value < 0.05 was required.

### Cell proliferation assay

Single-cell clones with MapUTR variants were seeded at 3,000 cells per well in the 96-well plates. For each single-cell clone, five technical replicates (wells) were performed. After 24 h incubation at 37°C, the plate was transferred to the Incucyte® S3 live-cell analysis system (Sartorius) to monitor cell proliferation. Images were taken every 2 h and analyzed for confluency. Data were analyzed and plotted using Graphpad Prism 7. *P*-values were calculated using one-tailed Student's *t* test. To call significance, *p*-value < 0.05 was required.

### RNA-seq analysis of prime-edited single-cell clones

For each gene, we randomly picked three single-cell clones per allele (except for the *IRAK1* alternative allele, which we only included two single-cell clones) and extracted total RNA for RNA-seq library generation at the UCLA Technology Center for Genomics and Bioinformatics (TCGB). The libraries were sequenced on Novaseq SP PE100. Reads were mapped to the human genome and transcriptome with HISAT2[99]. To identify differentially expressed (DE) genes between single-cell clones with the alternative allele and reference allele for each gene, we used DESeq2[100]. To call significance, we required cutoffs of $|\log_2 \text{FoldChange}| > 0.5$ and FDR < 0.1. For *IRAK1*, we used the top 500 DE genes ranked by FDR as query genes for GO enrichment analysis. For *MFN2* and *FOSL2*, all DE genes were included for GO enrichment analyses. For each query gene, a control gene was randomly chosen from the background genes (excluding all query genes) tested in DESeq2. We also required a control gene to match the query gene in gene length and GC content (±10%). To calculate the enrichment of a GO term in the query genes, we fit a normal distribution of the occurrences of the same GO term in 10,000 sets of control genes. A FDR < 0.05 and occurrence (number of genes associated with a term) ≥ 5 were required for significance. GO terms were ranked by $-\log_{10}(\text{FDR})$, and the top 15 GO terms were plotted.

### Reporting summary

Further information on research design is available in the Nature Portfolio Reporting Summary linked to this article.

## Data availability

The rare 3′ UTR variants tested in this study were obtained from gnomAD (https://gnomad.broadinstitute.org). The predicted miRNA target sites were obtained from the TargetScan database at https://www.targetscan.org/vert_72/vert_72_data_download/All_Target_Locations.hg19.bed.zip. The gene-disease association data was obtained from the DisGeNET database at https://www.disgenet.org/downloads. The Intogen data used in this study are available at https://intogen.org/download. The Cancer Genome Atlas (TGCA) data are available under restricted access adhere to the National Institutes of Health (NIH) Genomic Data Sharing (GDS) policy as well as the National Cancer Institute (NCI) GDS policy; FPKM values for cancer patients were obtained through the Genomic Data Commons (GDC) portal (https://portal.gdc.cancer.gov/); Genotype data were obtained through the ICGC Data Portal (http://dcc.icgc.org/pcawg/). Processed MapUTR data generated by this study is available at https://github.com/gxiaolab/mapUTR[101]. The GTEx data was obtained from the GTEx portal (https://www.gtexportal.org/). All raw MapUTR data have been deposited in NCBI's Gene Expression Omnibus and are accessible through GEO Series accession number GSE232573. Source data are provided with this paper.

## Code availability

Third-party tools were used for data analysis, such as UMItools (version 0.5.5), Botwtie2 (version 2.3.5), MPRAnalyze (https://github.com/YosefLab/MPRAnalyze), HOMER (verson 4.11), and DeepRiPe (https://github.com/ohlerlab/DeepRiPe). Scripts to use these tools are available upon request from the corresponding author.

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

## Acknowledgements

The authors thank members of the Xiao laboratory for helpful discussions and comments on this work. We thank Peter Freese and Chris Burge for sharing the RNA Bind-n-Seq data. The results published here are in part based upon data generated by The Cancer Genome Atlas managed by the National Cancer Institute (NCI) and National Human Genome Research Institute (NHGRI). This work was supported in part by grants from the National Institutes of Health (U01HG009417, R01CA262686, and R01AG075206 to X.X.) and the Jonsson Comprehensive Cancer Center at UCLA. T.F. was supported by the UCLA Hyde Fellowship and Dissertation Year Fellowship. K.A. was supported by the University of California-Historically Black Colleges and Universities (UC-HBCU) Fellowship. T.W.C. was supported by the NIH T32LM012424. S.T. was supported by the NIH T32GM145388. The content is solely the responsibility of the authors and does not necessarily represent the official views of the National Institutes of Health.

## Author contributions

T.F., K.A. and X.X. designed the study with inputs from all other authors. T.F., J.H.B., S.T. and R.C. conducted the molecular, cellular, and biochemical experiments. K.A., T.F., T.W.C. and J.H.L. conducted the bioinformatics analyses. X.X. and S.K. provided supervisory inputs. All authors contributed to the writing of the paper. All authors approved the final manuscript.

## Competing interests

The authors declare no competing interests.
