## [Peer Review File · Nature Communications]

Massively parallel screen uncovers many rare 3' UTR variants regulating mRNA abundance of cancer driver genesREVIEWER COMMENTS

Reviewer #1 (Remarks to the Author):

GWAS studies have shown that majority of disease-associated variants are in the non-coding regions of the genome. Functional interpretation of these associations is key to our understanding of the complex disease molecular etiology but remains challenging. The vast majority of genetic variants are rare variants, whose functional role in complex diseases remain poorly understood. Fu et al. developed a massively parallel reporter assay, MapUTR, to systematically identify the rare variants in the 3'UTR that may affect the mRNA abundance. They applied this method in two cell lines to screen 17,301 rare gnomAD variants and found an average of 24.5% could affect mRNA abundance. They further applied MapUTR to systematically interrogate 11,929 somatic mutations and uncovered 3,928 functional mutations in 155 cancer driver genes. In addition, they found that untranslated tumor mutational burden (uTMB), a metric reflecting the amount of somatic functional MapUTR variants of a tumor can be used as a prognostic marker and is associated with patient survival. Finally, using prime editing, they demonstrated that the functional variants in the cancer-relevant genes not only alter mRNA abundance, but also affect cell growth. Overall, it is a very interesting and important study that was well-designed/executed. My detailed comments are listed as follows.

1. The authors analyzed the effect of variants on miRNA targeting, or RBP binding motifs. However, the effect of the variants on RBP binding sites in the 3'UTR can be beyond known motifs. It would be informative to analyze how many functional variants reside in the RBP binding sites and their spatial distribution around the RBP binding motifs (if applicable) using the ENCODE RBP eCLIP-seq datasets.
2. It remains unclear whether the functional variants identified are uniformly distributed across 3'UTR or show some spatial preferences across the 3'UTR (e.g. near stop codon, polyA site or the middle of 3'UTR etc.).
3. A related question to #2: are there any functional variants that may overlap with the regions/motifs potentially affecting the alternative polyadenylation site usage?
4. For a gene with different isoforms, a given variant can be shared by different isoforms under different sequence contexts. Do the identified variants show consistent effect on different isoforms or different effects between isoforms when the sequence contexts are different?
5. As this study focuses on the rare variants, it would be interesting to compare the effect size of the identified functional variants with the effective size of the common variants identified in previous studies if the information about the effective size from previous studies is available?

6. What is the co-occurrence pattern of the identified variants from the same 3'UTR of cancer driver genes? For example, how many of them co-occur in the same patients? As the screen was performed in only two cell lines that cover limited cancer types, a discussion about the limitation of the current study on identifying cancer type-specific functional variants would be important.

7. The finding of an enrichment of functional rare variants in cancer-relevant genes could be due to the fact that both cell lines used for the massively parallel reporter assay are cell lines that undergo active proliferation. It would be important to discuss the limitations of using proliferating cell lines for interrogating the variant function because some of the functional variants may play a role in post-mitotic tissues such as neurons. It is also important to discuss potential approaches to expand the screen to post-mitotic cells.

8. Prime editing may have off-target effects that affect genes beyond the intended targets. It would be important to perform rescue experiments with moderate knockdown/overexpression of the intended targets (just to restore the expression level similar to that of the wild-type) to rule out the possibility that the observed phenotypes were confounded by the off-target effect.

Reviewer #2 (Remarks to the Author):

The paper describes the development of a method for screening functional variants in the 5' UTR, termed "MapUTR".

The authors have effectively utilized this technique to detect 6598 functional 5'UTR mutations, providing new insights into a yet-to-be-thoroughly investigated class of variants.

However, I have identified several points of concern within the manuscript, which I believe the authors need to address to improve the quality and integrity of their work.

- Since the authors performed screening based on rare "germline" variants, there often lacks detailed observation as to whether each functional variant works as germline or somatic.
- The manuscript lacks an in-depth discussion of the relationship between the functional variants in the 5'UTR and their effects on tumor suppressor genes (should decrease expression) or oncogenes (should increase expression).
- Often, there was no mention whether "functional variants" can be found in cancer genomes and in which types of cancer.

This is a significant gap in the manuscript, as a thorough analysis of such variants could provide valuable insights.

In several instances, the authors present arguments involving numerical data without establishing appropriate controls. This could potentially affect the validity of their results.

I have identified specific points that need to be addressed in the manuscript, mainly concerning the interpretation of the results.

Page 6:

- The correlation shown in Fig. 2e includes only functional variants. How would this change when all the tested mutations are taken into account?

- The authors should provide more information about the relationship between the allele frequency (AF) and the proportion of functional variants. Is there a trend that functional variants are concentrated in the low AF region?

- For some variants, confirmation should be possible using GTEx WGS and RNA-seq data. Does the RNA-seq data for each tissue, for variants with mutations in GTEx WGS, show changes in the expected direction? Is there any tissue heterogeneity?

- Do the sequence characteristics differ for functional variants that increase and those that decrease expression values?

- Functional MapUTR variants account for 63.8% and 63.5% respectively, and they match predicted miRNA target sites. Compared to the total number of variants experimented with, is this enriched?

- The target site of miR-34b-3p is highlighted, but when you look at each miRNA target site, is there anything like a particular site having high enrichment?

- Are the PLIN4 (rs767768172) and LDHD mutations (rs145078776) registered in disease databases like ClinVar? LDHD is said to be involved in clear cell renal carcinomas, but is there a significant mutation actually seen in ccRCC?

Page 7:

- "4,648 (70.4%) functional rare variants located in genes associated with cancer." the discussion ultimately proceeds with the union of the MapUTR variants of HEK293 and HeLa. Each had a False Discovery Rate (FDR) of 10%, but it seems to me that taking the union would considerably increase the FDR. Is this not a problem? Also, what percentage of the overall mutations were in cancer genes? Are there enrichment compared to overall 5'UTR used for experiments?

- Tumor suppressors are expected to be down-regulated. Rather than saying there's a tendency for one gene to be like this, can't it be shown as an overall trend?

Page 8:

- When testing 11,929 COSMIC mutations, it would be beneficial to compare the results with controls.
- In Fig 4e, the authors claim that "the overall effects of their functional variants were detrimental". Is this claim statistically significant?

Page 9:

- It would be insightful to know how many matches were found between the TCGA data from the PCAWG database and the functional variants identified in this study.
- A total of 113 functional variants were associated with patients whose gene expression value was an outlier (Fig. 5a). How significant is this association?
- In Fig 5a, CDKN2A, NF1, and ATM are tumor suppressors, and their effects should be caused by inactivation. However, in this figure, expression is increasing due to mutation. How should we interpret this?

Page 10:

- For the mutations being experimented with here (Fig. 6), can the authors confirm whether they are observed in the PCAWG or other databases?

Reviewer #3 (Remarks to the Author):

The manuscript entitled "Massively parallel screen uncovers many rare 3' UTR variants regulating mRNA abundance of cancer driver genes" by Ting Fu et al. reports the results of a massively parallel screen aiming to quantify the impact of rare single nucleotide variants on the stability of mRNAs. In principle, it is an interesting data set, given that the function of non-coding regions remains insufficiently characterized. However, due to the lack of some controls, the interpretation of the results is problematic for the reasons detailed below.

Major issues:

In general, the study focuses on rare variants in 3'UTRs. Given that non-coding regions evolve more rapidly than coding regions, the fact that these are rare variants could already be indicative of selection

(i.e. mutations are not tolerated in these specific non-coding regions). This could explain their functionality and make it less surprising that so many of them appear to have some functionality (even though the effect sizes seem to be very small, see Fig. 3a). Thus, the authors should appropriately frame the problem and not overstate their results. A second main issue is that often the relevant controls are missing, as mentioned in further comments. In particular, it is unclear whether the fraction of functional variants overlapping with some feature of the genome is any different than the fraction of all the rare variants (functional or not).

p. 4. "A DNA-seq library was also generated from the plasmid library (pre-transfection) to allow for normalizations of RNA expression"

The DNA level measurement is important for the normalization and the choice of the authors to quantify the DNA level abundance pre-transfection is very surprising because it makes the electroporation efficiency a strong confounding factor for all downstream analysis. This is why other MPRA studies measure the DNA level post-transfection see, e.g., here:

PMID: 24633241

PMID: 33046894

PMID: 35315433

This calls into question all conclusions based on these data. The authors should perform post-transfected DNA-seq and redo the downstream analysis or present convincing arguments on why this issue has negligible effects.

p. 4. "...we tested different DNA/Cell ratios and observed that 200ng DNA per 1M cells yielded similar RNA/DNA ratios as previously reported (Supplementary Table 1)"

In the supplementary table 1 it is clearly seen that the ratios oscillate in unexpected ways, the RNA/DNA ratio does not increase monotonically with the amount of DNA/Cell. It is hard to conclude anything from these numbers. Thus, this type of experiment should be repeated first in several replicates and then with higher concentrations to be able to argue that the selected concentration is indeed optimal.

p. 17-18 Estimation of variant effect sizes.

A main issue with this analysis is that the noise is higher for low RNA/DNA abundance measurements than for higher values. This is not taken into account in the analysis (for e.g. to calculate a maximum

likelihood value of the ratio) and could be the reason why there are such large fluctuations in the analysis of known motifs (Fig. 1f). Quantile normalization alone, applied by the authors, does not solve the problem. At the minimum, the authors should continue their analysis with sites for which there is enough data so that the sampling fluctuations in the reads cannot completely mask the changes.

p. 5 “A relatively low error rate (average 0.016%) was again observed in the DNA-seq and RNA-seq reads (Fig. 2a).”

As I understand, average mismatch rate in Fig. 2a means that the authors calculated mismatch rate for every position in every designed sequence in every replicate, and then took the average over all designed sequences and replicates in each position.

But what is the distribution of those mismatch rates? Are there designed sequences with particularly high mismatch rates? Probably they should be excluded from the analysis? Also, given that what is compared is the effect of different alleles, it is essential that in a pair of designed sequences ref allele - alt allele, there are no positions with high mismatch rates in the close vicinity.

p. 6. Consistent with this expectation, in both HEK293 and HeLa cells, we observed a significant bias toward upregulation of mRNA abundance by the alternative alleles in MapUTR (Fig. 3a).

This is one of the examples where a crucial control is missing. The relative scores should be compared between variants overlapping miRNA binding sites and variants that don't overlap any miRNA binding site. Moreover, the effects are extremely small and in the HEK293 cell line barely significant (a p-value of 0.002 with these many data points is disconcertingly large).

p. 6, 7, 18, 19 Motif discovery and motif strength analysis

Here again a control is missing. Motifs discovered in the comparison of upregulating sequence group with downregulating sequence groups (and vice versa), should be searched for in the sequences of non-functional variants, to get the control. Such a control should be used, for Fig. S3e, Fig 3c,d. Currently presented results do not provide any evidence that the change in motif strength is responsible for the observed relative score.

Minor issues:

p. 4. This reporter gene is driven by a strong promoter, the CMV early enhancer/chicken beta actin (CAG) promoter, which allows identification of functional

variants primarily affecting post-transcriptional (rather than transcriptional) regulation.

CMV is indeed a strong promoter. But the authors have no control on the transfection efficiency. So the argument they make based on the use of the CMV promoter cannot be made.

p. 4. "Our data demonstrate the likely existence of abundant non-coding cancer driver mutations."

This conclusion is overstated. There is a series of previous papers reporting many non-coding cancer driver mutations. See, e.g., here: <https://www.nature.com/articles/s41586-020-1965-x>, <https://www.nature.com/articles/s41568-021-00371-z>

The authors should specify more clearly what new biological insights about non-coding mutations in cancer driver genes their study brings, rather than make sweeping and vague statements.

p. 6 In HEK293 and HeLa cells, 63.8% and 63.5% of all functional MapUTR variants overlapped predicted miRNA target sites, respectively.

There is no description about how miRNA target sites were predicted. What was the density of sites, what sort of scores did they have etc. Furthermore, what fraction of all tested variants overlapped miRNA target sites?

p. 6 "Thus, the genetic background, rather than trans-acting factors, plays a dominant role in determining the function of many 3' UTR variants"

This is another example of a misleading interpretation of the results. In fact, 70% of variants that are functional in one cell line are not functional in the other cell line. The authors focus on the minority that behave similarly between the lines. Moreover they use vague terminology like "many variants" to overstate the results.

p.14. All oligo were included in chip1.

"chip1" is referred to before the actual definition.

p. 14. Based on GENCODE 86 basic v24 annotation, we selected...

This is one of the earliest versions of the GENCODE annotation for hg38 (from 2015), thus long outdated. What was the reason for this choice? At the minimum, the authors should intersect the set of tested variants with the latest version (v 43) to see if it yields the same set of 17,301 variants including 1,044 clinically relevant SNPs.

Figures:

Fig. 1 a - this seems to be not a “general” workflow, rather an “experimental scheme” or “experimental design”. Also, providing the definition of abbreviations would help (as is done, e.g., in panel c).

Fig 1 e,f - please include the explanation of the motif abbreviations in the legend/caption

Fig 1.e, Fig S2, a - why are the position coordinates not aligned with the tested region coordinates? I would expect position 1 to correspond to the 1st nucleotide of the tested region. I understand that other parts include primer binding sites and restriction enzyme site, but these are not described in the figure. Please either include these additional parts in Fig 1.d or remove those from Fig 1.e and Fig S2,a.

Fig. 1.f Fig S2,b - the authors may consider merging those two panels into one and indicating different cell lines with different colors. The replicability of the same pattern of relative activity scores in different cell lines is an important result.

Fig. 3b - wrong name of the miRNA that targets LDHD. It should be miR-3180-5p.

We thank all reviewers for their very thorough comments and constructive suggestions, which have helped strengthen our manuscript substantially. We have revised the manuscript, with the main changes highlighted in blue font. Below we provide a detailed point-by-point response.

REVIEWER COMMENTS

Reviewer #1 (Remarks to the Author):

GWAS studies have shown that majority of disease-associated variants are in the non-coding regions of the genome. Functional interpretation of these associations is key to our understanding of the complex disease molecular etiology but remains challenging. The vast majority of genetic variants are rare variants, whose functional role in complex diseases remain poorly understood. Fu et al. developed a massively parallel reporter assay, MapUTR, to systematically identify the rare variants in the 3'UTR that may affect the mRNA abundance. They applied this method in two cell lines to screen 17,301 rare gnomAD variants and found an average of 24.5% could affect mRNA abundance. They further applied MapUTR to systematically interrogate 11,929 somatic mutations and uncovered 3,928 functional mutations in 155 cancer driver genes. In addition, they found that untranslated tumor mutational burden (uTMB), a metric reflecting the amount of somatic functional MapUTR variants of a tumor can be used as a prognostic marker and is associated with patient survival. Finally, using prime editing, they demonstrated that the functional variants in the cancer-relevant genes not only alter mRNA abundance, but also affect cell growth. Overall, it is a very interesting and important study that was well-designed/executed. My detailed comments are listed as follows.

We appreciate the reviewer's thorough summary of our work and the recognition of the importance and the well-designed nature of our study. We thank the reviewer for raising the thoughtful questions below and have addressed these questions and updated our manuscript accordingly.

1. The authors analyzed the effect of variants on miRNA targeting, or RBP binding motifs. However, the effect of the variants on RBP binding sites in the 3'UTR can be beyond known motifs. It would be informative to analyze how many functional variants reside in the RBP binding sites and their spatial distribution around the RBP binding motifs (if applicable) using the ENCODE RBP eCLIP-seq datasets.

We examined the overlap between our functional variants and ENCODE RBP eCLIP peaks from K562 or HepG2 cells. 57.6% or 57.5% of functional variants in HEK293 (2,195) or HeLa (2,701), respectively, reside within the binding peak of at least one RBP. To determine the spatial distribution of functional variants around RBP binding sites, we calculated the distance between each variant and the closest binding peak for each RBP according to ENCODE eCLIP data (Van Nostrand, Eric L et al. 2020; PMID 32728246). We designated a distance of zero to variants that overlap a binding peak. The distribution of functional variants around RBP binding peaks demonstrated a higher density within and closer to the peaks, compared to nonfunctional variants (Fig. R1). We have included this analysis as Fig. S7 (page 7).

Fig. R1: The spatial distribution of functional and nonfunctional variants around RBP binding sites according to ENCODE eCLIP data. P-value was calculated using the Kolmogorov-Smirnov test.

2. It remains unclear whether the functional variants identified are uniformly distributed across 3'UTR or show some spatial preferences across the 3'UTR (e.g. near stop codon, polyA site or the middle of 3'UTR etc.).

To address this question, we examined the number of functional variants in decile bins along the length of the 3' UTRs. Fig. R2 shows the proportion of functional variants among all tested variants in each decile. Overall, the functional variants are relatively uniformly distributed across the length of the 3' UTR (KS-test goodness of fit: $p = 0.998$). We have included this figure as Fig. S6a, and updated the text accordingly (page 6).

Fig. R2: Proportion of tested variants that are functional in various regions along the length of the 3' UTR. The regional distribution is not different from a uniform distribution (KS-test goodness of fit $p = 0.998$).

3. A related question to #2: are there any functional variants that may overlap with the regions/motifs potentially affecting the alternative polyadenylation site usage?

To address the reviewer's question, we checked whether any functional variants overlapped a polyA signal (AAUAAA/AUUAAA) or an UGUA motif while requiring the UGUA to be upstream and within 70nt of any polyA signal (Cheng Y, et al. 2006. PMID: 16870936; Tian B, et al. 2012. PMID: 22012871). Among all functional variants detected in the gnomAD and COSMIC datasets, we identified 34 unique variants overlapping a polyA signal. Also, 16 of the 34 variants overlapping a polyA signal did not have an upstream UGUA motif within 70nt, thus may not be functional. The 18 remaining variants make up 0.10%, 0.28% and 0.08% of all functional variants identified in gnomAD (HEK293), gnomAD (HeLa) and COSMIC respectively. Thus, the

effect of our functional variants on alternative polyadenylation is expected to be small. The MapUTR assay is not designed to capture such variants. However, it can potentially be modified to detect variant effects on alternative polyadenylation in the future. We updated the Discussion to address this point (page 15). This question also reminded us to update Fig. S2 to illustrate the location of the pre-designed polyA signal and UGUA motif outside of the designed oligo libraries in the plasmid.

4. For a gene with different isoforms, a given variant can be shared by different isoforms under different sequence contexts. Do the identified variants show consistent effect on different isoforms or different effects between isoforms when the sequence contexts are different?

We agree it would be ideal to test each variant in its natural sequence context at the isoform level. However, such experiments require long synthetic sequences to capture isoform differences in the reporter. Current MPRA methods are limited by the length of synthetic oligos (up to 300nt) that are available *en masse*. Thus, this work did not aim to resolve context-specific function of a given variant. Only one isoform sequence context was used for each variant (see Methods, page 16). Future endeavors, such as CRISPR editing-based screens combined with long-read sequencing, may be helpful in addressing this question. We have updated the Discussion to include this limitation (page 15).

5. As this study focuses on the rare variants, it would be interesting to compare the effect size of the identified functional variants with the effective size of the common variants identified in previous studies if the information about the effective size from previous studies is available?

It is an interesting question to compare the effect sizes of common and rare variants. Due to differences in selection criteria of the tested variants, experimental approaches and computational analyses, the effect sizes resulted from previous studies and ours may not be readily comparable. Thus, we conducted another MapUTR experiment to test 3,367 common 3' UTR variants in HeLa cells. Please note that these common variants were predicted as functional variants as part of a parallel ongoing study in our lab. A total of 1,200 (35.6%) of these common variants were detected as functional using the same computational analysis and significance cutoffs (minimum of 10% fold change and FDR ≤ 0.1). Note that this percentage (35.6%) is relatively high as the tested variants were selected for their potential functionality. Fig. R3 below shows the effect size comparison. Indeed, rare variants had significantly higher effect sizes compared to common variants. This is consistent with the expectation that functional rare variants are generally more disruptive than common ones. We have added this new result as Fig. S6b and updated the text (page 6).

Fig. R3: Absolute effect sizes of functional rare and common variants tested with MapUTR in HeLa cells. The p -value was calculated using the Kolmogorov–Smirnov test.

6. What is the co-occurrence pattern of the identified variants from the same 3'UTR of cancer driver genes? For example, how many of them co-occur in the same patients? As the screen was performed in only two cell lines that cover limited cancer types, a discussion about the limitation of the current study on identifying cancer type-specific functional variants would be important.

To address this question, we searched for functional MapUTR variants in cancer driver genes among TCGA patients of different cancer types. In total, 405 functional variants (in 105 cancer driver genes) were found in patient genotype data. We then examined the number of co-occurring variants in each gene in each patient (Fig. R4a). A small number of genes (e.g., *RET*, *RECQL4*, *CUX1*, *FGFR4*, *PIK3R1*, *FGFR3*, and *GNAS*) had some co-occurring variants per patient in different cancer types. The vast majority of genes did not harbor co-occurring variants. For genes with at least 2 variants co-occurring in one patient, we further asked whether these variants induce mRNA abundance changes in the same direction (Fig. R4b). Most genes had co-occurring variants functional in opposite directions in at least one patient. However, interestingly, variants co-occurred in the oncogenes *FGFR3*, *CCND1*, and *MYC* mostly increased mRNA abundance. In contrast, those in the tumor suppressor gene *PIK3R1* and *ATM* mostly decreased mRNA abundance. Thus, co-occurrence of these detrimental variants (pro-cancer) may exhibit synergistic effects on mRNA abundance regulation, which should be tested in future studies. We have added the above results to the manuscript (page 9, Fig. S12).

We agree with the reviewer about the limitation of cancer types in our study and have discussed this point in the Discussion (page 15).

Fig. R4: Co-occurrence pattern of functional MapUTR cancer somatic mutations in TCGA patients. a, Heatmap of the number of co-occurred variants in the same cancer driver gene (column) in each TCGA patient (row). b, Number of times a pair of functional MapUTR cancer somatic mutations were observed in the same gene across all patients. Genes with at least two functional variants co-occurred in one patient are plotted. TSG, tumor suppressor gene. Opposite direction, the co-occurred variants had opposite effect on mRNA abundance (one increase, one decrease). Both decrease, both of the co-occurred variants decreased mRNA abundance. Both increase, both of the co-occurred variants increased mRNA abundance.

7. The finding of an enrichment of functional rare variants in cancer-relevant genes could be due to the fact that both cell lines used for the massively parallel reporter assay are cell lines that undergo active proliferation. It would be important to discuss the limitations of using proliferating cell lines for interrogating the variant function because some of the functional variants may play a role in post-mitotic tissues such as neurons. It is also important to discuss potential approaches to expand the screen to post-mitotic cells.

We have updated the Discussion to address the limitations of using proliferating cells, and how MapUTR may be applied to post-mitotic cells (page 15).

8. Prime editing may have off-target effects that affect genes beyond the intended targets. It would be important to perform rescue experiments with moderate knockdown/overexpression of the intended targets (just to restore the expression level similar to that of the wild-type) to rule out the possibility that the observed phenotypes were confounded by the off-target effect.

Prime editing has been shown to induce much lower off-target editing than Cas9 in previous studies (Anzalone AV, et al. 2019. PMID: 31634902; Chen, Peter J et al. 2021. PMID: 34653350). To reduce the potential bias due to off-target effects, we have isolated at least 4 single-cell clones for each reference and alternative allele after prime editing perturbation. As suggested by the reviewer, we have tried to perform rescue experiments in single-cell clones with alternative alleles. However, in order to restore the expression level of genes and observe the cellular phenotype, we needed to generate stable cell lines using lentivirus, which by itself decreased the cell proliferation of the transduced cells. We were not able to rescue the cell proliferation rate in these lentivirus-treated cells, probably because the rescue effect was masked by the effects of lentiviral transduction. As an alternative approach, we did Sanger sequencing to check the top three potential off-target sites predicted by CRISPRitz (Cancellieri, Samuele et al. 2020. PMID: 31764961), for each epegRNA or nick sgRNA used in this study (Supplementary Table 2). We did not detect any off-target editing in these sites (Appendix 1). Thus, we believe that the observed phenotypes in cell proliferation were unlikely to be confounded by the off-target effects. We have updated the text to address this point (page 12).

Reviewer #2 (Remarks to the Author):

The paper describes the development of a method for screening functional variants in the 5' UTR, termed "MapUTR".

The authors have effectively utilized this technique to detect 6598 functional 5'UTR mutations, providing new insights into a yet-to-be-thoroughly investigated class of variants.

However, I have identified several points of concern within the manuscript, which I believe the authors need to address to improve the quality and integrity of their work.

We thank the reviewer for a thorough review of our manuscript. We appreciate the questions and concerns raised by the reviewer and have addressed them in detail below.

- Since the authors performed screening based on rare "germline" variants, there often lacks detailed observation as to whether each functional variant works as germline or somatic.

Common to all MPRA experiments, the sequence of interest (variant in this case) and a short flanking region are cloned into a reporter. In this artificial context, any variant, either germline or somatic, can be tested. However, whether the functional variant works as a germline or somatic variant will depend on its in vivo context, that is, the genotype of the human individual. In addition, a germline variant in one individual could be a somatic variant in another. Thus, the same variant could function as a germline or somatic variant in a specific context, and it is hard to generally label a variant as functional germline or somatic variant. We have updated the manuscript to clearly state the source of the variants (either from a germline or somatic database).

- The manuscript lacks an in-depth discussion of the relationship between the functional variants in the 5'UTR and their effects on tumor suppressor genes (should decrease expression) or oncogenes (should increase expression).

We clarified in detail the relationship between the 3' UTR functional variants and their effects on cancer driver genes (page 9). Specifically, 56 of the 62 (90%) tumor suppressor genes had at least one variant that attenuates mRNA abundance, while 43 of the 50 (86%) oncogenes harbored at least one variant that enhances mRNA abundance. Yet not all 3'UTR functional variants found in oncogenes would increase gene expression and vice versa for tumor suppressor genes. These functional variants may be considered benign variants in tumors. Nonetheless, even a single functional mutation may cause detrimental effect by enhancing oncogene expression or repressing expression of a tumor suppressor. Please refer to page 9 for more details.

- Often, there was no mention whether "functional variants" can be found in cancer genomes and in which types of cancer. This is a significant gap in the manuscript, as a thorough analysis of such variants could provide valuable insights.

We agree this is an important question. In our analysis of the functional variants in TCGA data, we have found 2,853 functional variants in cancer patients (both germline and somatic) across 22 cancer types in TCGA. A total of 508 out of 2,853 functional variants were associated with gene expression outliers. We have updated Table S6 to include all functional variants found in cancer patients in TCGA. In addition, responding to Reviewer 1's comment, we have expanded the cancer genome-related analysis (please see our response to Reviewer 1 #6). We agree that such analyses yielded valuable insights.

In several instances, the authors present arguments involving numerical data without establishing appropriate controls. This could potentially affect the validity of their results.

I have identified specific points that need to be addressed in the manuscript, mainly concerning the interpretation of the results.

Page 6:

- The correlation shown in Fig. 2e includes only functional variants. How would this change when all the tested mutations are taken into account?

Fig. R5 shows the correlation of relative activity scores of all tested mutations between the two cell lines, which is still significant. This level of correlation (R value) is weaker than that between functional variants (Fig. 2e), which is expected as functional variants should have larger and more robust effect sizes than insignificant variants. In the original manuscript, we highlighted the functional variants because they are of higher interest than the nonfunctional ones. Nonetheless, we have now included Fig. R5 as Fig. S6c and updated the text accordingly (page 6)

Fig. R5: Spearman correlation of the relative activity of all tested rare variants between HEK293 and HeLa cells.

- The authors should provide more information about the relationship between the allele frequency (AF) and the proportion of functional variants. Is there a trend that functional variants are concentrated in the low AF region?

To address this question, we binned the MapUTR tested gnomAD variants into 7 groups based on their reported allele frequency (AF). We did not observe a trend of a higher proportion of functional variants in low AF bins (Fig. R6). This may be because the AFs of rare variants are already very low (<1%) and further grouping them into small bins did not make a difference.

Fig. R6: Proportion of functional gnomAD variants in different allele frequency bins.

- For some variants, confirmation should be possible using GTEx WGS and RNA-seq data. Does the RNA-seq data for each tissue, for variants with mutations in GTEx WGS, show changes in the expected direction? Is there any tissue heterogeneity?

As suggested by the reviewer, we identified GTEx individuals with a MapUTR functional variant and compared the expression level of their host gene to the expression in individuals without the variant. This analysis was performed for each gene in each tissue with the goal of identifying gene expression outliers using the same method as for the TCGA data in the manuscript. Across the 2,251 rare MapUTR functional variants that exist in at least 1 GTEx individual, 352 showed outlier expression and consistent direction of change as detected by the MapUTR assay. Examples are shown in Fig. R7a below. Please note that we do not expect all functional rare variants to be associated with outlier expression because the regulation of each gene's expression level is intricate that involves many cis-elements and trans-factors.

To determine whether the outlier gene expression status was heterogeneous across tissues, we plotted a heatmap of all gene expression outliers across all tissues (Fig. R7b). We observed that for each gene only a handful of tissues showed outlier status, indicating that there is heterogeneity in the variant effect as similarly noted by other studies (Arvanitis, Marios et al. 2022. PMID: 35085493; Sul, Jae Hoon et al. 2013. PMID: 23785294).

We have included the above results as Fig. S15 and updated the text accordingly (page 11).

Fig. R7: Functional MapUTR variants in GTEx tissues. **a**, Examples of MapUTR variants with outlier expression in GTEx tissues. **b**, Heatmap of all outlier genes (columns) in all tissues (rows). For each gene, if an outlier was found in at least one sample of the tissue, we plotted the gene as dark red (upregulating) or dark blue (downregulating). Otherwise, we plotted the mean z-score of the gene's FPKM in samples with functional variants.

- Do the sequence characteristics differ for functional variants that increase and those that decrease expression values?

In the motif analysis, we differentiated between functional variants that increase versus decrease expression. This analysis revealed distinct sets of motifs associated with downregulation or upregulation of the host gene (Fig. S8). To further address the reviewer's question, we analyzed features of the designed sequences containing downregulating and upregulating variants by comparing their mononucleotide and dinucleotide frequencies and GC content. However, we did not observe significant difference between these general sequence characteristics. Thus, we kept the motif analysis results in the manuscript.

- Functional MapUTR variants account for 63.8% and 63.5% respectively, and they match predicted miRNA target sites. Compared to the total number of variants experimented with, is this enriched?

Compared to the total number of variants tested, the functional SNPs were not significantly enriched with miRNA target sites. This lack of significance may be due to the difficulty in predicting miRNA target sites computationally. The next question raised by the reviewer motivated us to do the analysis for individual miRNAs. This analysis identified hundreds of miRNAs whose targets were enriched with functional variants (see below). Although it still suffers from limited accuracy of miRNA target predictions, this analysis shows disrupting miRNA target sites may be one functional mode of 3' UTR variants. We have updated the manuscript to clarify the above points (page 7).

- The target site of miR-34b-3p is highlighted, but when you look at each miRNA target site, is there anything like a particular site having high enrichment?

We highlight miR-34b-3p as a positive example of variant effects on mRNA abundance potentially through miRNA targeting. To identify miRNAs whose targets are highly enriched among functional variants, for each miRNA, we calculated the proportion of variants disrupting the miRNA target sites among all functional variants. A similar proportion was calculated among nonfunctional variants and a Fisher's exact test was performed for significance. We identified 726 and 988 miRNAs in HEK293 and HeLa, respectively, that are significantly enriched among functional variants. In Fig. R8, we present the top 20 miRNAs from HEK293 (a) and HeLa (b). Some miRNAs such as miR-328-5p and miR-6885-5p consistently showed up among the top miRNAs, suggesting the potential commonality in miRNA target disruption between the two cell lines. We have updated the manuscript to clarify the above points (page 7).

Fig. R8: Top 20 miRNAs overlapping with functional variants more than nonfunctional variants in **a**, HEK293 and **b**, HeLa.

- Are the *PLIN4* (rs767768172) and *LDHD* mutations (rs145078776) registered in disease databases like ClinVar? *LDHD* is said to be involved in clear cell renal carcinomas, but is there a significant mutation actually seen in ccRCC?

We did not find the *PLIN4* (rs767768172) or *LDHD* (rs145078776) mutations in disease databases such as ClinVar (Landrum MJ, et al. 2018. PMID: 29165669), CIViC (Griffith, Malachi et al., 2017. PMID: 28138153), COSMIC (Forbes SA, et al. 2008. PMID: 18428421), or iGAP (Ruiz, A et al. 2014. PMID: 24495969). As the reviewer suggested, a previous study showed that low *LDHD* expression was associated with worse overall survival in ccRCC patients (Wang, Yue et al. 2018. PMID: 29963157). In MapUTR, we identified two functional variants (rs145078776 and rs372014702) in *LDHD*, both increased mRNA abundance of *LDHD*. We did not observe these two *LDHD* variants in clear cell renal carcinoma patients in the TCGA analysis. We chose to highlight these two genes in the manuscript due to the clear miRNA targeting relevance of their variants, which may not have been cataloged in existing studies or databases.

- "4,648 (70.4%) functional rare variants located in genes associated with cancer." The discussion ultimately proceeds with the union of the MapUTR variants of HEK293 and HeLa. Each had a False Discovery Rate (FDR) of 10%, but it seems to me that taking the union would considerably increase the FDR. Is this not a problem?

The reviewer makes a good point. However, we want to clarify that this sentence simply aims to provide the observed number of total functional rare variants, which by no means was a statistical statement. We thought the readers would be interested in knowing this number after combining both cell lines. Such statements were commonly made in previous studies, e.g., MPRAu (Griesemer, Dustin et al. 2021. PMID: 34534445), where the union of functional variants across six cell lines was mentioned.

Also, what percentage of the overall mutations were in cancer genes? Are there enrichment compared to overall 5'UTR used for experiments?

The percentage of all tested gnomAD rare variants in cancer-related genes is 71.1% (12,312 out of 17,312). We did not observe an enrichment of functional gnomAD variants in cancer-related genes compared to the overall 3' UTR gnomAD variants tested in MapUTR. Thus, it is an interesting note that, in general, gnomAD rare variants frequently occur in cancer-related genes, which is consistent with the expectation that rare variants are often detrimental and selected against, and the fact that we observed a larger fraction of functional variants among the tested rare variants, compared to common variants (see response to Reviewer 1 #5). We have updated the manuscript to reflect this point (page 14).

- Tumor suppressors are expected to be down-regulated. Rather than saying there's a tendency for one gene to be like this, can't it be shown as an overall trend?

We agree with the reviewer that tumor suppressors are expected to be down-regulated by a detrimental functional mutation in cancer. However, a gene may harbor different types of variants (detrimental, neutral or even beneficial to cancer). Indeed, we observed functional variants that alter tumor suppressor expression in either direction (Fig. 4e). In disease, it is possible that one detrimental mutation drives critical disease processes, despite the existence of many other mutations. Thus, an overall trend may not show the expected pro-cancer direction. Among the cancer driver genes we tested, 56 of the 62 (90%) tumor suppressor genes had at least one variant that attenuates mRNA abundance, while 43 of the 50 (86%) oncogenes harbored at least one variant that enhances mRNA abundance. Thus, pro-cancer detrimental mutations exist in many genes, but each gene may also harbor non-detrimental mutations. We have updated the text to clarify this point (page 9).

Page 8:

- When testing 11,929 COSMIC mutations, it would be beneficial to compare the results with controls.

In MapUTR, we tested both the reference allele and the alternative allele for each COSMIC mutation, with the reference allele serving as control. We calculated a relative activity score by comparing the mutant allele with the reference allele. Functional mutations were called based on the relative activity score and FDR cutoff (See Methods). Thus, each mutant allele had its own internal control. If the reviewer is referring to control variants relative to rare COSMIC mutations, the new common variant MapUTR testing could serve as such a control (please see response to Reviewer 1 #5).

- In Fig 4e, the authors claim that “the overall effects of their functional variants were detrimental”. Is this claim statistically significant?

We thank the reviewer for this question. This statement was not meant to be a statistical one. To enhance clarity, we have revised the text (page 9) as follows. “Note that a functional variant can be detrimental (pro-cancer), neutral or beneficial in tumors depending on its effect on mRNA abundance and the role of the cancer driver genes (tumor suppressing or oncogenic). It is possible that one detrimental mutation drive critical disease processes, despite the existence of many non-detrimental mutations.”

In addition, we replaced Fig. 4e with violin plots (also shown below as Fig. R9) which more clearly show the specific data points and their scores.

Fig. R9: Distribution of relative activity scores for oncogenes and tumor suppressor genes.

Page 9:

- It would be insightful to know how many matches were found between the TCGA data from the PCAWG database and the functional variants identified in this study.

We found 2,853 functional variants in TCGA patients across 22 cancer types. We have updated Table S6 to include all functional variants detected in cancer patients in TCGA.

- A total of 113 functional variants were associated with patients whose gene expression value was an outlier (Fig. 5a). How significant is this association?

Compared to non-functional variants, this level of association was not significant. It should be noted that the existence of outlier association is supportive of the functional roles of a variant. Nonetheless, the absence of such an association does not indicate non-functionality of a variant. Since gene expression is regulated via multiple types of mechanisms, multiple genetic variants could contribute to the regulation of one gene. Thus, it is possible that the function of one mutation is not manifested as a single driver of gene expression. We have updated the Discussion to reflect this point (page 15).

- In Fig 5a, CDKN2A, NF1, and ATM are tumor suppressors, and their effects should be caused by inactivation. However, in this figure, expression is increasing due to mutation. How should we interpret this?

This question is related to a question above. We again note that functional variants detected by MapUTR are not necessarily detrimental to the patients. For the three examples shown in Fig 5a, the functional variants in tumor suppressor genes *CDKN2A*, *NF1*, and *ATM* increased gene expression, indicating that these variants may not be detrimental to cancer patients. Multiple types of mutations (detrimental, neutral or beneficial) may exist in a gene, but it may only take one detrimental mutation to drive a cancer phenotype.

Page 10:

- For the mutations being experimented with here (Fig. 6), can the authors confirm whether they are observed in the PCAWG or other databases?

We did not observe these three variants (*MFN2*: rs777822288, *FOSL2*: rs11884725, and *IRAK1*: rs782486025) in the PCAWG or other clinically relevant databases, including ClinVar (Landrum MJ, et al. 2018. PMID: 29165669), CIViC (Griffith, Malachi et al.. 2017. PMID: 28138153), COSMIC (Forbes SA, et al. 2008. PMID: 18428421), and iGAP (Ruiz, A et al. 2014. PMID: 24495969). Nonetheless, since the 3 genes have interesting cancer-relevant functions and these variants affects mRNA abundance and cellular proliferation, we think they are still functionally important despite their absence in current databases.

Reviewer #3 (Remarks to the Author):

The manuscript entitled “Massively parallel screen uncovers many rare 3’ UTR variants regulating mRNA abundance of cancer driver genes” by Ting Fu et al. reports the results of a massively parallel screen aiming to quantify the impact of rare single nucleotide variants on the stability of mRNAs. In principle, it is an interesting data set, given that the function of non-coding regions remains insufficiently characterized. However, due to the lack of some controls, the interpretation of the results is problematic for the reasons detailed below.

We thank the reviewer for a thorough review of our manuscript. We appreciate that the reviewer recognizes that the study/data set is interesting. We have clarified about the controls used in our analyses and carried out additional analyses as suggested by the reviewer to address all the concerns.

Major issues:

In general, the study focuses on rare variants in 3’UTRs. Given that non-coding regions evolve more rapidly than coding regions, the fact that these are rare variants could already be indicative of selection (i.e. mutations are not tolerated in these specific non-coding regions). This could explain their functionality and make it less surprising that so many of them appear to have some functionality (even though the effect sizes seem to be very small, see Fig. 3a). Thus, the authors should appropriately frame the problem and not overstate their results.

We thank the reviewer for this comment. We have revised the Introduction and Discussion of the manuscript as suggested (pages 4, 13).

A second main issue is that often the relevant controls are missing, as mentioned in further comments. In particular, it is unclear whether the fraction of functional variants overlapping with some feature of the genome is any different than the fraction of all the rare variants (functional or not).

We’ve addressed the specific questions below.

p. 4. “A DNA-seq library was also generated from the plasmid library (pre-transfection) to allow for normalizations of RNA expression”

The DNA level measurement is important for the normalization and the choice of the authors to quantify the DNA level abundance pre-transfection is very surprising because it makes the electroporation efficiency a strong confounding factor for all downstream analysis. This is why other MPRA studies measure the DNA level post-transfection see, e.g., here:

PMID: 24633241

PMID: 33046894

PMID: 35315433

This calls into question all conclusions based on these data. The authors should perform post-transfected DNA-seq and redo the downstream analysis or present convincing arguments on why this issue has negligible effects.

This is an important question. Indeed, we generated both pre- and post-transfection DNA libraries when we developed MapUTR (we apologize for not including these comparisons in the original manuscript). While post-transfection DNA-seq may overcome the bias associated with transfection efficiency, it could also introduce new noises due to the low recovery rate (< 5%) of plasmids from transfected cells. The latter problem is potentially severe in our assay as we transfect 5-20 fold less plasmids than in previous studies to avoid exhaustion of cellular machineries. In the process of developing the MapUTR protocol, we collected both pre- and post-transfection DNA-seq libraries with 3 biological replicates each, incorporating 15-mer unique molecular identifiers (UMIs) in library generation (Fig. S2b). Following sequencing, we calculated the alternative allele frequency (AF) ($\text{alt}/(\text{alt}+\text{ref})$) for each variant using the UMI counts from the two types of DNA-seq libraries. Indeed, their AFs are highly correlated (Fig. R10a), especially at high UMI abundances, although the correlation is still significant given low UMI counts (1st quantile) (Fig. R10b-c). Given the ease of generating pre-transfection DNA libraries, we opted to use pre-transfection DNA to perform RNA abundance normalization.

We appreciate the list of publications that utilized post-transfection DNA measurement in MPRA studies. It should be noted that these studies mostly used lentivirus-mediated stable transfection methods. Cells were cultured for longer times (~3 days) (Klein, Jason C et al. 2020. PMID: 33046894; Kreimer, Anat et al. 2022. PMID: 35315433), or passaged several times to remove residual lentiviral RNA (Zhao, Wenxue et al. 2014. PMID: 24633241). The use of post-transfection DNA would be appropriate in these cases since the composition of the library may change due to random genome integrations or the prolonged cell culture that causes gene dropouts. MapUTR is a transient reporter assay, in which cells were harvested 24h after transfection. We have seen several MPRA studies, such as Griesemer, Dustin et al. 2021. PMID: 34534445, Cooper, Yonatan A et al. 2022. PMID: 35981026, and Mattioli, Kaia et al. 2020. PMID: 32819422, that employed a similar transient transfection strategy as ours and used pre-transfection plasmid DNA for normalization.

We have included the above data as Fig. S3 and updated the text accordingly (pages 4-5).

Fig. R10. Comparison between post- and pre-transfected DNA for RNA normalization. **a**, Spearman correlation of alternative allele frequencies (alt/(alt+ref)) between pre- and post-transfected DNA. **b**, Scatter plot of mean total counts (alt + ref) between pre- and post-transfected DNA separated into 4 quantiles. The 1st quantile represents the group of variants with the least amount of UMI counts. The 4th quantile represents the highest UMI counts. **c**, Spearman correlation of alternative allele frequencies (alt/(alt+ref)) between pre- and post-transfected DNA in different quantiles defined in **b**.

p. 4. "...we tested different DNA/Cell ratios and observed that 200ng DNA per 1M cells yielded similar RNA/DNA ratios as previously reported (Supplementary Table 1)"

In the supplementary table 1 it is clearly seen that the ratios oscillate in unexpected ways, the RNA/DNA ratio does not increase monotonically with the amount of DNA/Cell. It is hard to conclude anything from these numbers. Thus, this type of experiment should be repeated first in several replicates and then with higher concentrations to be able to argue that the selected concentration is indeed optimal.

We have repeated this optimization experiment with higher DNA/Cell inputs (4ug/1M), as well as with three biological replicates for each condition. We did not repeat the previous positive controls as we realized that they had additional variants close to the ones we tested. Instead, we selected three known functional variants in the literature (Zhao, Wenxue et al. 2014. PMID: 24633241; Adams, Brian D et al. 2007. PMID: 17312270), and measured their impacts given three DNA/Cell ratios respectively. The largest ratio, 4ug DNA per 1 million cells (4ug/1M), was chosen as it was used in previous literature (Mattioli, Kaia et al. 2020. PMID: 32819422) and the smallest ratio, 40ng/1M, was chosen because ratios lower than this level did not permit the recovery of the RNA libraries. We observed that the smaller DNA/Cell ratio yielded larger effect sizes, in line with the expected directions of RNA/DNA ratios (Fig. R11). We hypothesized that this increased sensitivity may be due to the low DNA input, which avoids exhaustion of the cellular regulatory machineries. However, we found that the targeted RNA-seq library had very

low yield (<1ng) at 40ng/1M DNA/Cell ratio. Thus, we opted to use a DNA/Cell ratio (200ng/1M) that is as low as possible while yielding expected directions of RNA/DNA ratios (Fig. R11) and reliable generation of RNA libraries (10-180ng). Compared to other studies (Griesemer, Dustin et al. 2021. PMID: 34534445; Mattioli, Kaia et al. 2020. PMID: 32819422; Cooper, Yonatan A et al. 2022. PMID: 35981026), we were able to lower the DNA/Cell input ratio by 5-20 fold.

We have included the above data as Fig. S1 and update the text accordingly (page 4).

Fig. R11: DNA/Cell ratio optimization using cell electroporation. **a**, A list of control SNPs used for optimization tests. **b**, RNA/DNA ratios observed for the alternative allele of the control variants when transfected under different DNA (ng)/Cell (M) ratios. Three biological replicates were included in each experiment. Data are plotted as mean ± SEM.

p. 17-18 Estimation of variant effect sizes.

A main issue with this analysis is that the noise is higher for low RNA/DNA abundance measurements than for higher values. This is not taken into account in the analysis (for e.g. to calculate a maximum likelihood value of the ratio) and could be the reason why there are such large fluctuations in the analysis of known motifs (Fig. 1f). Quantile normalization alone, applied by the authors, does not solve the problem. At the minimum, the authors should continue their analysis with sites for which there is enough data so that the sampling fluctuations in the reads cannot completely mask the changes.

We agree that low counts will have higher noise in estimating sequence activity. But we would like to note that we obtained high coverage per designed sequence across 3 biological replicates. In fact, among the sequences tested in Fig. 1f, the lowest read count observed for any sequence in any replicate was 305. The fluctuations in relative activity scores of known motifs may be due to the fact that we tested all possible base mutations in each position in each motif and these are graphed collectively in Fig. 1f. As different nucleotides may have different functional impacts at each position, it is not surprising that the graphs showed fluctuations. To further address this concern, we examined the read coverage in the rare variant datasets. The proportion of all tested variants with less than 25 reads per replicate is very small (0.67%), but their RNA/DNA ratios are still correlated across replicates Fig. R12). Moreover, we used the

MPRAnalyze method (Ashuach, T. et al. 2019, PMID: 31477158) to detect functional variants, which uses a maximum likelihood estimation to optimize the underlying linear model for differential activity. Therefore, we expect low counts do not severely affect our results.

Fig. R12: Correlation of RNA/DNA ratios for sequences with low coverage (<25 reads) sequence.

p. 5 “A relatively low error rate (average 0.016%) was again observed in the DNA-seq and RNA-seq reads (Fig. 2a).”

As I understand, average mismatch rate in Fig. 2a means that the authors calculated mismatch rate for every position in every designed sequence in every replicate, and then took the average over all designed sequences and replicates in each position.

But what is the distribution of those mismatch rates? Are there designed sequences with particularly high mismatch rates? Probably they should be excluded from the analysis? Also, given that what is compared is the effect of different alleles, it is essential that in a pair of designed sequences ref allele – alt allele, there are no positions with high mismatch rates in the close vicinity.

As the reviewer pointed out, Fig. 2a shows the average mismatch rate for every designed sequence in each position. As suggested by the reviewer, we analyzed the mismatch rate for each designed sequence separately. One subpool (4,000 sequences) is shown below for illustration purpose (Data from other subpools are similar). First, we examined the average mismatch rate across the length of each designed sequence. As shown in Fig. R13, this mismatch rate per sequence was generally very low, with a large proportion (40%) of designed sequences having nearly no mismatches (Fig. R13 left). Further, the average mismatch rate for the 20 bases (+/-10nt) flanking the designed SNP was also low (Fig. R13 right) and lower than that in the full-length sequence, possibly because these SNPs were mostly located at the center of the synthetic oligoes by design.

Examining the mismatches in each designed sequence, we observed that in only 1.4% (56) of the designed sequences, at least one base position had a >10% mismatch rate, and 0.73% (29) had at least 2 base positions with >10% mismatch rate. We further asked how many designed sequences had high mismatch rates (>5%, per base) in the close vicinity of the designed SNP (+/- 5nt). Only 0.15% (6) of the designed sequences passed these requirements. Overall, we opted to keep these designed sequences given their small number. We have included the above results in the manuscript (Fig. S5, page 6).

Fig. R13: Mismatch rates of designed sequences. Distribution of mismatch rates across the full length (left) and in the local vicinity (+/- 10nt, right) of the designed sequences.

p. 6. Consistent with this expectation, in both HEK293 and HeLa cells, we observed a significant bias toward upregulation of mRNA abundance by the alternative alleles in MapUTR (Fig. 3a).

This is one of the examples where a crucial control is missing. The relative scores should be compared between variants overlapping miRNA binding sites and variants that don't overlap any miRNA binding site. Moreover, the effects are extremely small and in the HEK293 cell line barely significant (a p-value of 0.002 with these many data points is disconcertingly large).

We agree with the reviewer that it would be good to use other variants as controls. However, after considering this question more carefully, we realized that comparison of relative scores between functional variants overlapping miRNA target sites and functional variants that do not overlap any miRNA target sites may not be productive. miRNA targeting is only one of multiple potential mechanisms through which a variant may affect RNA abundance, e.g., changes in RBP binding may also lead to upregulation. Therefore, we do not have a strong reason to expect that functional variants that do not overlap miRNAs are not upregulated. Also, we considered to use nonfunctional variants as controls. However, since, by definition, these variants have small relative activity scores centered around zero, this comparison is not meaningful. Thus, we decided to do the analysis by testing whether the mean relative activity score is greater than zero, internally controlled by the presence of downregulating variants (against upregulating variants). The shift towards positive scores is consistent with the expected direction of miRNA targeting disruption. We agree that the effects are extremely small, which may partially reflect the challenging nature of *in silico* miRNA target predictions. Despite this, we do note that certain miRNA target sites (Fig. 3b showing examples) may be strongly affected by MapUTR variants. As suggested by Reviewer 2, we asked whether functional MapUTR variants tend to disrupt certain miRNAs more often than nonfunctional variants. This analysis identified hundreds of such miRNAs (726 and 988 in HEK293 and HeLa respectively). Although it still suffers from limited accuracy of miRNA target predictions, this analysis shows disrupting miRNA target sites may be one functional role of 3' UTR variants. We have updated the text (page 7) to interpret the results with caution.

p. 6, 7, 18, 19 Motif discovery and motif strength analysis

Here again a control is missing. Motifs discovered in the comparison of upregulating sequence group with downregulating sequence groups (and vice versa), should be searched for in the sequences of non-functional variants, to get the control. Such a control should be used, for Fig. S3e, Fig 3c,d. Currently presented results do not provide any evidence that the change in motif strength is responsible for the observed relative score.

We initially used random dbSNPs or shuffled motifs as controls. As suggested by the reviewer, we randomly sampled the same number of nonfunctional variants as the functional variants and scored them against the motifs. Unlike the functional ones, the nonfunctional variants did not disrupt the detected motifs. In fact the changes in motif strength caused by the functional variants were significantly higher compared to those for the nonfunctional and shuffled PWM controls (Fig. R14, now Fig. S8e). We also included the nonfunctional variants as controls for Fig.3c, 3d and S10b. Our conclusions remain unchanged, in that the functional variants caused higher RBP binding changes than nonfunctional variants (Fig. R15). We have updated the text accordingly (pages 7-8).

Fig. R14: Motif strength is significantly altered by functional MapUTR variants compared to nonfunctional ($p < 2.2e-16$) and shuffled PWM controls ($p < 2.2e-16$). P-values were calculated using Kolmogorov-Smirnov test.

Fig. R15: Functional MapUTR variants significantly alter RBP binding. The p-value indicated in the plot is the maximum p-value obtained from two comparisons (functional versus nonfunctional variants, and functional versus random dbSNPs). P-values were calculated using Kolmogorov-Smirnov test.

Minor issues:

p. 4. This reporter gene is driven by a strong promoter, the CMV early enhancer/chicken beta actin (CAG) promoter, which allows identification of functional variants primarily affecting post-transcriptional (rather than transcriptional) regulation.

CMV is indeed a strong promoter. But the authors have no control on the transfection efficiency. So the argument they make based on the use of the CMV promoter cannot be made.

In a typical MapUTR experiment, 1.5ug of plasmids containing both reference alleles and alternative alleles were transfected simultaneously into the cells. Unlike traditional minigene experiments which transfect the reference and alternative alleles into cells separately, MapUTR transfects reference and alternative alleles into the same pool of cells. We hypothesized that the transfection process should be highly random and should not introduce a specific bias towards a certain allele. As we have shown above (question 3 from reviewer 3), the alternative allele frequency is significantly correlated between pre- and post-transfection DNA libraries, indicating that our assay is robust to transfection and plasmid isolation. We have updated the main text to address this concern (page 5).

p. 4. "Our data demonstrate the likely existence of abundant non-coding cancer driver mutations."

This conclusion is overstated. There is a series of previous papers reporting many non-coding cancer driver mutations. See, e.g., here: <https://www.nature.com/articles/s41586-020-1965-x>, <https://www.nature.com/articles/s41568-021-00371-z>

The authors should specify more clearly what new biological insights about non-coding mutations in cancer driver genes their study brings, rather than make sweeping and vague statements.

We have updated this text to be more specific (page 4). We note that previous studies mostly relied on the mutation rate and regulatory element annotations to screen for non-coding cancer driver mutations (Rheinbay, Esther et al. 2020. PMID: 32025015). In this study, we focused on functional 3' UTR cancer mutations independent of regulatory element annotations. Our data demonstrated the likely existence of abundant 3' UTR cancer mutations in cancer driver genes.

p. 6 In HEK293 and HeLa cells, 63.8% and 63.5% of all functional MapUTR variants overlapped predicted miRNA target sites, respectively.

There is no description about how miRNA target sites were predicted. What was the density of sites, what sort of scores did they have etc. Furthermore, what fraction of all tested variants overlapped miRNA target sites?

We thank the reviewer for bringing this to our attention. We have included details about our miRNA analysis under the Methods section (pages 20-21). Briefly, the miRNA target site data was obtained from TargetScan predictions and were filtered for high quality (context++ percentile ≥ 50). On average, the genes with favorable sites (defined by TargetScan for high-confidence sites) contained a median of 386 miRNA target sites. Next, we overlapped functional MapUTR variants with the predicted target sites, requiring the functional SNP to be located within the genomic region of the miRNA seed. 64.4% of all tested variants overlapped miRNA target sites and there is no significant difference between this proportion and that observed for

functional variants. This lack of significance may again be due to the difficulty in predicting miRNA target sites computationally. We have updated the manuscript to clarify the above points (page 7).

p. 6 “Thus, the genetic background, rather than trans-acting factors, plays a dominant role in determining the function of many 3’ UTR variants”

This is another example of a misleading interpretation of the results. In fact, 70% of variants that are functional in one cell line are not functional in the other cell line. The authors focus on the minority that behave similarly between the lines. Moreover they use vague terminology like “many variants” to overstate the results.

We understand this concern. First, we added a Fig. S6c to show the comparison of the relative activity scores of all tested variants between the two cell lines (Fig. R5 above). There exists a significant positive correlation among all tested variants in the two cell lines, although the majority of variants are functional in only one cell line. Whether a variant was detected as functional or not depends on many factors, such as the read coverage levels and the cutoffs used to call functional variants. When focusing on only the functional variants shared by the two cell lines, the correlation of relative activity scores (Fig. 2e) was stronger than shown in Fig. R5. Thus, the overall data support concordance of variant function between the two cell lines. Nonetheless, we do understand the reviewer’s concern and updated the text to be more specific and explicit in stating this conclusion (page 6).

p.14. All oligo were included in chip1.
“chip1” is referred to before the actual definition.

Thank you for the comment. We named different batches of oligo pools “chips”. To be clear, we have replaced the word “chip” with “oligo pool” to avoid confusion.

p. 14. Based on GENCODE 86 basic v24 annotation, we selected...
This is one of the earliest versions of the GENCODE annotation for hg38 (from 2015), thus long outdated. What was the reason for this choice? At the minimum, the authors should intersect the set of tested variants with the latest version (v 43) to see if it yields the same set of 17,301 variants including 1,044 clinically relevant SNPs.

We used GENCODE v24 because it was an existing up-to-date version when we started our project. We have intersected all tested gnomAD variants with the current GENCODE release 44 (available in 07.2023). 17,085 of the 17,301 tested gnomAD variants, and 1,022 of the 1,044 clinically relevant SNPs, remain as 3’ UTR variants based on GENCODE v44. For the variants (216 of all tested including 22 of the clinically relevant ones) that are not in annotated 3’ UTRs of GENCODE v44, 212 (including 22 clinically relevant ones) are indeed still annotated as 3’ UTR variants by NCBI RefSeq or UCSC/ Ensembl Genes. Among the 6,598 functional variants, 6,595 are still annotated as 3’ UTR variants. Given the small number of differences, we opted to retain the results based on the v24 annotation.

Figures:

Fig. 1 a - this seems to be not a “general” workflow, rather an “experimental scheme” or “experimental design”. Also, providing the definition of abbreviations would help (as is done, e.g., in panel c).

We have updated the figure legend as suggested.

Fig 1 e,f - please include the explanation of the motif abbreviations in the legend/caption

We have updated the legend of Fig. 1 and Fig. S4 to include the explanation of the motif abbreviations.

Fig 1.e, Fig S2, a - why are the position coordinates not aligned with the tested region coordinates? I would expect position 1 to correspond to the 1st nucleotide of the tested region. I understand that other parts include primer binding sites and restriction enzyme site, but these are not described in the figure. Please either include these additional parts in Fig 1.d or remove those from Fig 1.e and Fig S2,a.

Thank you for this suggestion. We modified Fig. 1e and Fig. S4 to exclude the primer sequences. The updated figures now show the actual genomic coordinates and match the structure of the tested region in Fig. 1d. Note that the mutated region surrounding the SAMD4A motif is much closer to the end of the 3' UTR compared to the other motifs based on the GENCODE annotation, resulting in an off-centered mutated region. This is because the tested motif is close to the boundary of the annotated 3' UTR of its host gene *CHRD1*; thus the tested region did not extend beyond the 3' UTR.

Fig. 1.f Fig S2,b - the authors may consider merging those two panels into one and indicating different cell lines with different colors. The replicability of the same pattern of relative activity scores in different cell lines is an important result.

As suggested, we merged the two panels into one. The direction of relative activity is largely consistent between the two cell lines, despite some differences in the magnitude. Such differences in magnitude may be due to potential cell-type differences that influence the effect of a variant in the 3' UTR.

Fig. 3b - wrong name of the miRNA that targets LDHD. It should be miR-3180-5p.

We thank the reviewer for pointing out this error in naming. This has been fixed.

REVIEWERS' COMMENTS

Reviewer #1 (Remarks to the Author):

The authors have satisfactorily addressed my comments. I have no further comments.

Reviewer #2 (Remarks to the Author):

The author has made several improvements in this revision. However, the reviewer feels that it still does not sufficiently address the concerns previously raised. This study focused on narrowing down functional variants in 3' UTRs already presumed functional (Zhao et al., Nature Biotechnology, 2014) using a massive parallel reporter assay. Therefore, compared to the 17,312 mutations selected by Zhao et al., those demonstrated as functional in this study should have been a more concentrated representation of various important genomic functions.

> Across the 2,251 rare MapUTR functional variants that exist in at least 1 GTEx individual, 352 showed outlier expression and consistent direction of change as detected by the MapUTR assay.

Firstly, there is no description of the total number of RNA-seq in GTEx. While the criteria for outliers are not overly lenient, the figure of 352 out of 2,251 does not seem particularly high when looking across various tissues. The author should have demonstrated that the "functional" variants had a significantly higher number of outliers compared to the original Zhao's variant set.

> The percentage of all tested gnomAD rare variants in cancer-related genes is 71.1% (12,312 out of 17,312). We did not observe an enrichment of functional gnomAD variants in cancer-related genes compared to the overall 3' UTR gnomAD variants tested in MapUTR.

This too seems quite weak. The reviewer is concerned that this may indicate that the current experiment has not sufficiently concentrated on the important elements compared to the original data (Zhao et al.).

> Among the cancer driver genes we tested, 56 of the 62 (90%) tumor suppressor genes had at least one variant that attenuates mRNA abundance, while 43 of the 50 (86%) oncogenes harbored at least one variant that enhances mRNA abundance.

The reviewer thinks this approach is flawed. They assume that the 62 tumor suppressors have numerous MapUTR variants, so it's possible that a single variant might be downregulated by chance, without leading to overall down-regulation. It was necessary to show that the MapUTRs associated with tumor suppressor genes are downregulated as a whole.

Presenting examples where typical tumor suppressors like CDKN2A, NF1, ATM (Figure 5a) are overexpressed is problematic.

In each gene, there are specific types of cancer that are associated. For example, APC is specifically seen in colorectal cancer. This perspective is completely lacking in the paper. It is inconceivable to demonstrate an example of over-expression of APC, which is thought to be inactivated in cancer, in the context of prostate cancer, especially since it is a tumor suppressor.

Despite preparing a list of cancer-relevant genes, MFN2, FOSL2, and IPAK1 highlighted in Fig 6 are not on this list. Nevertheless, these are being presented as mutations in cancer driver genes. Then claiming in the Abstract, "We characterized three variants in cancer-relevant genes (MFN2, FOSL2, and IPAK1)," seems to lack scientific integrity.

Overall, the author does not believe this paper is suitable for publication in Nature Communication.

Reviewer #4 (Remarks to the Author):

All concerns have been addressed.

Thanks for reaching out regarding reviewer #2. To be honest, I don't think reviewer #2 understood this study or has given it a serious read. In the original comments, this reviewer thought the author studied 5' UTR variants in all of their comments, but the study focused solely on 3' UTR. Most of the new concerns this reviewer raised are due to misunderstandings. First, they thought this study was testing a subset of variants already tested by Zhao et al 2014 (ref 25). This is false because these two studies are totally unrelated. The authors here tested an entirely new set of 3' UTR variants. They have described it clearly in the main text. Another misunderstanding is that the reviewer thought the functional 3' UTR variants are driving tumorigenesis, which is not what the author claimed (they simply claimed it's affecting cancer gene expression. It could affect tumor either way).

Reviewer #2 (Remarks to the authors)

The author has made several improvements in this revision. However, the reviewer feels that it still does not sufficiently address the concerns previously raised. **This study focused on narrowing down functional variants in 3' UTRs already presumed functional (Zhao et al., Nature Biotechnology, 2014) using a massive parallel reporter assay.** Therefore, compared to the 17,312 mutations selected by Zhao et al., those demonstrated as functional in this study should have been a more concentrated representation of various important genomic functions.

Again, the reviewer misunderstood the design of this study. The variants tested here are rare variants in gnomad, unrelated to Zhao et al 2014.

> Across the 2,251 rare MapUTR functional variants that exist in at least 1 GTEx individual, 352 showed outlier expression and consistent direction of change as detected by the MapUTR assay.

Firstly, there is no description of the total number of RNA-seq in GTEx. While the criteria for outliers are not overly lenient, the figure of 352 out of 2,251 does not seem particularly high when looking across various tissues. **The author should have demonstrated that the "functional" variants had a significantly higher number of outliers compared to the original Zhao's variant set.**

Again this study has nothing to do with Zhao's variants. Also the problem with GTEx data is that the genetic background is very heterogeneous. The individual with a particular variant of interest could have many other variants in the promoter/splice site/5' UTR/coding region that affect the same gene. The control, which is everyone else with the reference allele in GTEx, have many more variants in the same gene. That's the limitation of GTEx data. It's real human data but too much heterogeneity and very complicated. To get a robust signal, one will need a much larger cohort with hundreds of individuals carrying the rare allele, which simply doesn't exist.

> The percentage of all tested gnomAD rare variants in cancer-related genes is 71.1% (12,312 out of 17,312). We did not observe an enrichment of functional gnomAD variants in cancer-related genes compared to the overall 3' UTR gnomAD variants tested in MapUTR.

This too seems quite weak. The reviewer is concerned that this may indicate that the current experiment has not sufficiently concentrated on the important elements compared to the original data (Zhao et al.).

Again this study has nothing to do with Zhao et al. As the authors noted, the high percentage of rare gnomAD variants in cancer related genes is interesting but does not diminish the significance of this study. Those not identified to be functional in this study could be functional via mechanisms other than post-transcriptional regulation.

> Among the cancer driver genes we tested, 56 of the 62 (90%) tumor suppressor genes had at least one variant that attenuates mRNA abundance, while 43 of the 50 (86%) oncogenes harbored at least one variant that enhances mRNA abundance.

The reviewer thinks this approach is flawed. They assume that the 62 tumor suppressors have numerous MapUTR variants, so it's possible that a single variant might be downregulated by chance, without leading to overall down-regulation. It was necessary to show that the MapUTRs associated with tumor suppressor genes are downregulated as a whole.

As the authors have stated clearly, 'functional' means impact on mRNA abundance, rather than driving tumorigenesis as the reviewer apparently has in mind. Some of these variants could inhibit tumor growth by downregulating oncogenes or upregulating tumor suppressor genes.

Presenting examples where typical tumor suppressors like CDKN2A, NF1, ATM (Figure 5a) are overexpressed is problematic.

Fig 5a simply shows that some variants result in higher expression compared to reference allele. It's not saying that tumors over-express these tumor suppressor genes.

In each gene, there are specific types of cancer that are associated. For example, APC is specifically seen in colorectal cancer. This perspective is completely lacking in the paper. It is inconceivable to demonstrate an example of over-expression of APC, which is thought to be inactivated in cancer, in the context of prostate cancer, especially since it is a tumor suppressor.

Again the PRAD_APC panel in fig 5a simply shows that the 3' UTR SNP is associated with higher expression of APC. It has nothing to do with whether APC is overexpressed in cancer or not.

Despite preparing a list of cancer-relevant genes, MFN2, FOSL2, and IPAK1 highlighted in Fig 6 are not on this list. Nevertheless, these are being presented as mutations in cancer driver genes. Then claiming in the Abstract, "We characterized three variants in cancer-relevant genes (MFN2, FOSL2, and IPAK1)," seems to lack scientific integrity.

I honestly don't know what list the reviewer is talking about. I don't see any problem listing these 3 genes as cancer relevant genes, especially after the experiments in this study. Even if these genes are not in some list, why is this a scientific integrity issue?